

# Impact of Spaceborne Carbon Monoxide Observations from the S-5P platform on Tropospheric Composition Analyses and Forecasts

R. Abida[1], J.-L. Attié[1,2], L. El Amraoui[1], P. Ricaud[1], W. Lahoz[3], H. Eskes[4], A. Segers[5], L. Curier[5], J. de Haan[4], J. Kujanpää[6], A. O. Nijhuis[4], D. Schuettemeyer[7], J. Tamminen[6], R. Timmermans[5], P. Veefkind[4], and B. Veihelmann[7]

[1] CNRM-GAME, Météo-France/CNRS UMR 3589, Toulouse, France

[2] Université de Toulouse, Laboratoire d'Aérologie, CNRS UMR 5560, Toulouse, France

[3] NILU – Norwegian Institute for Air Research, P.O. Box 100, 2027 Kjeller, Norway

[4] Royal Netherlands Meteorological Institute (KNMI), P.O. Box 201, 3730 AE De Bilt, The Netherlands

[5] TNO, Business unit Environment, Health and Safety, P.O. Box 80015, 3508 TA Utrecht, The Netherlands

[6] Finnish Meteorological Institute, Earth Observation Unit, P.O. Box 503, 00101 Helsinki, Finland

[7] ESA/ESTEC, Earth Observation Programmes, Noordwijk, The Netherlands

Submitted to Atmospheric Chemistry and Physics Discussions, 13 November 2015





**Abstract**
We use the technique of Observing System Simulation Experiments (OSSEs) to quantify the impact of
spaceborne carbon monoxide (CO) total column observations from the Sentinel-5 Precursor (S-5P) platform
on tropospheric analyses and forecasts. We focus on Europe for the period of northern summer 2003, when
there was a severe heat wave episode associated with extremely hot and dry weather conditions. We describe
different elements of the OSSE: (i) the Nature Run (NR), i.e., the "Truth"; ii) the CO synthetic observations;
(iii) the assimilation run (AR), where we assimilate the observations of interest; (iv) the control run (CR), in
this study a free model run without assimilation; and (v) efforts to establish the fidelity of the OSSE results.
Comparison of the results from AR and the CR, against the NR, shows that CO total column observations
from S-5P provide a significant benefit (at the 99% confidence level) at the surface, with the largest benefit
occurring over land in remote regions. Furthermore, the S-5P CO total column observations are able to
capture phenomena such as the forest fires that occurred in Portugal during summer 2003. These results
provide evidence of the benefit of S-5P observations for monitoring processes contributing to atmospheric
pollution.





## 1. Introduction

Over the last decade, the capabilities of satellite instruments for sensing the lower troposphere have improved, and opened the way for monitoring and better understanding of atmospheric pollution processes, e.g., tropospheric chemistry (Jacob, 2000), long-range transport (HTAP, 2007), and emissions (e.g. Streets D., 2013 and references therein). Satellite instruments provide global measurements of many pollutants (e.g., ozone; carbon monoxide, CO; nitrogen dioxide, $NO_2$; and aerosols), including information on their trans-boundary transport, and complement in situ measurements from ground-based stations (e.g., the EMEP, http://www.nilu.no/projects/ccc/emepdata.html, and Airbase, http://acm.eionet.europa.eu/databases/airbase/, networks). Low Earth Orbit (LEO) satellite platforms have the advantage of providing observations with global coverage, but at a relatively low temporal resolution. Geostationary Earth Orbit (GEO) satellite platforms provide observations at a continental scale, i.e., not global, but at a much higher temporal resolution.

Satellite data, either in synergy with ground-based and airborne measurements and/or assimilated into models such as chemistry transport models (CTMs), contribute to an improved understanding of tropospheric chemistry and dynamics and improved forecasts of atmospheric pollutant fields (see, e.g., Elbern et al., 2010). As part of an integrated observing strategy, satellite measurements provide a global view on air quality (AQ). The challenge for future space-borne missions will be to assess directly the local scales of transport and/or chemistry for tropospheric pollutants (1 hour or less, 10 km or less) and to facilitate the use of remote sensing information for improving local- and regional-scale (from country-wide to continental scales) AQ analyses and forecasts. Building on this effort, various LEO satellite platforms and/or constellations of GEO satellite platforms will help extend AQ information from continental scales to global scales (e.g., Lahoz et al., 2012, and references therein for LEO/GEO platforms; Barré et al., 2015, for GEO platforms).

An atmospheric species of interest for monitoring AQ is CO, owing to its relatively long time-scale in the troposphere; its distribution provides information on the transport pathways of atmospheric pollutants. Spaceborne instruments on LEO satellite platforms demonstrate the potential of remote sensing from space



to determine the CO distribution and its main emission sources at the global scale (Edwards et al., 2004,
2006; Buchwitz et al., 2006; Worden et al., 2013 and references therein). These LEO satellite platforms
include MOPITT (Measurements Of Pollution In The Troposphere), IASI (Infrared Atmospheric Sounding
Interferometer), AIRS (Atmospheric InfraRed Sounder) and TES (Tropospheric Emission Spectrometer)
operating in the thermal infrared (TIR) and SCIAMACHY (SCanning Imaging Absorption spectroMeter for
Atmospheric ChartographY) operating in the short-wave infrared (SWIR), respectively. By contrast, to our
knowledge, there are no GEO satellite platforms measuring the CO distribution. However, despite their
potential, owing to limited revisit time, and relatively coarse spatial resolution, LEO instruments are not
optimal for monitoring regional and local aspects of air quality.

Copernicus is the current European Programme for the establishment of a European capability for Earth
Observation    (http://www.copernicus.eu/pages-principales/services/atmosphere-monitoring).    The    main
objective of the Copernicus Atmospheric Services is to provide information on atmospheric variables (e.g.,
the    essential    climate    variables,    ECVs;    https://www.wmo.int/pages/prog/gcos/index.php?name=
EssentialClimateVariables) in support of European policies regarding sustainable development and global
governance of the environment. The Copernicus Atmospheric Services cover: AQ, climate change/forcing,
stratospheric ozone and solar radiation. The services rely mainly on data from Earth Observation satellites.

To ensure operational provision of Earth Observation data, the space component of the Copernicus
programme includes a series of spaceborne missions developed and managed by the European Space Agency
(ESA) and EUMETSAT. Among them, three missions address atmospheric composition. These are the
Sentinel-5 (S-5) and Sentinel-5 Precursor (S-5P) from a LEO satellite platform, and the Sentinel-4 (S-4)
from a GEO satellite platform. The goal of the S-4 is to monitor key atmospheric pollutants (e.g., ozone;
$NO_2$; sulphur dioxide, $SO_2$; bromine monoxide, BrO; and formaldehyde) and aerosols at relatively high
spatio-temporal resolution over Europe and North Africa (8 km; 1 hour). The goal of the S-5 and S-5P
platforms is to provide global daily measurements of atmospheric pollutants (e.g., CO, ozone, $NO_2$, $SO_2$,
BrO, and formaldehyde), climate related trace gases (e.g., methane, $CH_4$) and aerosols, at relatively high
spatial resolution (from below 8 km to below 50 km, depending on wavelength).



The S-5P is the ESA pre-operational mission required to bridge the gap between the end of the OMI (Ozone
Monitoring Instrument) and the SCIAMACHY missions and the start of the S-5 mission planned for 2020
onwards. The S-5P scheduled launch is in 2016 with a 7 years design lifetime. The S-5P will fly in an early
afternoon sun-synchronous LEO geometry with an Equator crossing mean local solar time of 13:30, chosen
to allow the instrument to measure the strong pollution signal present in the afternoon. In contrast, the
GOME-2 (Global Ozone Monitoring Experiment - 2) platform collects data at a local solar time of 09:30
(when the pollution signal is relatively weak) and thus has a lower predictive value (Veefkind et al., 2012,
and references therein). The S-5P LEO platform will address the challenge of limited revisit time from LEOs
by providing unprecedented high spatial resolution of 7x7 km,  and improved sensitivity in the Planetary
Boundary Layer (PBL), allowing resolution of, e.g., derived CO emission sources at finer scales than
hitherto. The PBL varies in depth throughout the year, but is contained within the lowermost troposphere
(heights 0-3 km), and typically spans the heights 0-1 km.

A method to objectively determine the added value of future satellite observations such as S-4, S-5 and S-5P,
and to investigate the impact of different instrument designs, is that of Observing System Simulation
Experiments (OSSEs) commonly based on data assimilation (e.g., Lahoz and Schneider, 2014). The OSSEs
have been extensively used and shown to be useful in the meteorological community to test the impact of
future meteorological observations on the quality of weather forecasts (Nitta, 1975; Atlas, 1997; Lord et al.,
1997; Atlas et al., 2003). In a recent paper, Timmermans et al. (2015) review the application of OSSEs to
assess future missions to monitor AQ. The OSSEs are increasingly being used by the space agencies to assess
the added value of future instruments to be deployed as part of the Global Observing System (e.g., work on
the ESA Earth Explorer ADM-Aeolus; Tan et al., 2007).

Although the usefulness of OSSEs is well established, they have limitations, discussed in Masutani et al.
(2010a, b). A frequent criticism of OSSEs is that they are overoptimistic, largely owing to the difficulties of
representing the real Earth System (e.g., the atmosphere), even with state-of-the-art numerical models.
Nevertheless, even if overoptimistic, OSSEs provide bounds on the impact of new observing systems. For
example, if additional instruments provide no significant impact within an OSSE, they are unlikely to do so
in reality.

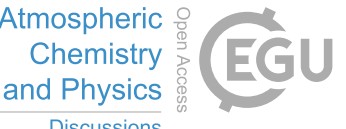




In this paper, we describe a regional-scale OSSE over Europe for northern summer 2003 (1 June – 31
August) to explore the impact of S-5P CO total column measurements on lowermost tropospheric air
pollution analyses, with a focus on CO PBL concentrations. The severe heat wave experienced in Europe
during northern summer 2003, and the concomitant atmospheric pollution and fire episodes, had a strongly
negative societal impact, being responsible for the deaths of over 14,000 people in France (Vautard et al.,
2005). This period had extremely hot and dry weather conditions and the long lasting atmospheric blocking
conditions significantly contributed to the accumulation of pollutants in the PBL owing to extended
residence time of the air parcels (Solberg et al., 2008). The spatial distribution of the enhanced levels of CO
and ozone was much more widespread over Europe during that summer than in previous ones (Lee et al.,
2006; Ordoñez et al., 2010). These exceptional weather conditions also resulted in several extreme wildfire
episodes over the Iberian Peninsula and the Mediterranean coast (Barbosa et al., 2004). Tressol et al. (2008)
point out that between 6 and 10 August 2003 the contribution of biomass burning to measured CO levels in
the lowermost troposphere reached 35% of the total CO field at these levels, a value comparable to typical
European anthropogenic emissions which represent 30% of this total CO field. Thus, the three-month period
1 June - 31 August 2003 includes both extreme and normal conditions, and provides an opportunity to study
the full range of pollution levels that occur in a summer season over Europe.

The OSSE study domain covers the larger part of Europe (5W-35E, 35N-70N), and we perform the OSSE
simulations at the spatial resolution of 0.2 degrees (latitude and longitude). This corresponds to a spatial
resolution of ~20 km (meridionally) and ~15 km (zonally, at 45N). With this spatial resolution, we can track
long-range transport plumes of CO. The length of the study period ensures we can sample different
meteorological situations typical for summertime, and provides an acceptable compromise between run-time
restrictions and provision of sufficient information for statistically significant results.

The structure of the paper is as follows. In Sect. 2 we describe the various components of the OSSE; in Sect.
3 we present the results from the OSSE for S-5P during summer 2003 over Europe. Finally, Sect. 4 provides
conclusions and identifies further work. A guiding principle in the OSSE set-up in this paper is to avoid
overoptimistic results.




## 2. The OSSE set-up

The OSSE concept consists of simulating observations and their associated errors from a representation of
reality (the "Nature Run" or NR) and providing this information to a data assimilation system to produce
estimates of the NR states. Thereafter, one compares these estimates of the NR states from an assimilation
run, AR (where the observation of interest has been assimilated), and from a control run, CR (in this case a
free model run), against the NR. The performance of the AR and the CR against the NR quantifies the benefit
of the observation of interest.

The OSSEs are widely used in the meteorological community for assessing the usefulness of new
meteorological satellite data. Recent examples (not exhaustive) include the work of Lahoz et al. (2005),
Stoffelen et al. (2006), and Tan et al. (2007); Masutani et al. (2010a) reviews the OSSE methodology and
provides a comprehensive list of references of OSSEs for meteorological applications. By contrast, there are
relatively few studies concerning OSSEs for AQ applications (Edwards et al., 2009; Timmermans et al.,
2009a, b; Claeyman et al., 2011; Zoogman et al., 2011; 2014a, b; Yumimoto, 2013). In a recent review,
Timmermans et al. (2015) comment that documented AQ OSSEs have demonstrated the benefits that could
accrue from proposed and planned satellite platforms for AQ monitoring and forecasting. In the study
described in this paper, the set-ups for the NR, and the CR and AR, use different models, thereby avoiding
the identical twin problem typically associated with overly optimistic OSSE results (see, e.g., Masutani et al.,
2010a). In Sects. 2.1-2.5 we describe the various elements of the OSSE study described in this paper. Figure
1 in Timmermans et al. (2015) provides a schematic showing the relationships between the various elements
in an OSSE.

### 2.1 The Nature Run

A key element of an OSSE is the NR that defines the true state used to evaluate analyses and/or forecasts
using simulated observations. The NR commonly consists of a long, free-running forecast evolving
continuously in a dynamically consistent way (Masutani et al. 2010a, b). For this study, the basis of the NR
consists of two high-resolution free model simulations performed with: (i) the regional LOTOS-EUROS air





quality model (Schaap et al., 2008), and (ii) the global chemistry transport model TM5 (Huijnen et al., 2010).
We obtain the NR by combining the LOTOS-EUROS CO profiles from the surface to 3.5 km with the TM5
CO profiles from 3.5 km to the top of the atmosphere (identified by the TM5 model top at 0.1 hPa). We use
spatial interpolation to merge the values near the boundary between the two models at a height of 3.5 km.
The model simulations used to construct the NR have a spin-up period of three months. We archive the NR
output data on an hourly basis.

To construct the NR, we run the LOTOS-EUROS model at a horizontal resolution of about 7 km nested into
the TM5 model, the latter run with a zoom domain over Europe at 1x1 degrees resolution. The TM5 model
has 34 layers with a model top at 0.1 hPa. The LOTOS-EUROS model describes air pollution in the
lowermost troposphere. It has four vertical layers following the dynamic mixing layer approach. The first
layer is a fixed surface layer of 25 metres thickness, the second layer (boundary layer) follows the mixing
layer height, and there are two reservoir layers spanning the rest of the atmosphere up to 3.5 km. The implicit
assumption of the LOTOS-EUROS model is the presence of a well-mixed boundary layer, so constituent
concentrations are constant up to the top of the Planetary Boundary Layer. The meteorological data used as
input for the LOTOS-EUROS model come from the European Centre for Medium-Range Weather Forecasts
(ECMWF). Prescription of surface anthropogenic emission is from the TNO-MACC-II emission database
(Kuenen et al., 2014), and fire emissions are from the MACC global fire assimilation system (GFAS v1;
Kaiser et al., 2012).

In the design of an OSSE, it is important to demonstrate that the NR exhibits the same statistical behaviour
as the real atmosphere in every aspect relevant to the observing system under study (Masutani et al., 2010a,
b). For the LOTOS-EUROS model used to build the lowermost levels of the NR, there is extensive
verification by comparison with European data and by frequent participation in international model
comparisons. This is the case for ozone and particulate matter (see Hass et al., 2003; Cuvelier et al., 2007;
van Loon et al., 2007; Stern et al., 2008; Manders et al., 2009; Curier et al., 2012; Márecal et al., 2015). To
evaluate the NR, we compare the surface CO data to available ground-based CO measurements over Europe
during northern summer 2003 (1 June – 31 August). For this comparison, we use the ground-based stations
from the Airbase database. We consider all types of ground-based stations from this database because of the


limited number of available measurements, but we discard stations with less than 75% of hourly data within
a month. This provides 171 ground-based stations for the comparison against the NR (note this approach
results in a paucity of stations over France).

Figure 1 shows the location of the selected Airbase ground-based stations measuring CO over Europe during
northern summer 2003 (top panel), and the time-series of CO concentrations during 1 June – 31 August
2003, measured by the selected Airbase ground-based stations and simulated by the NR and the CR (bottom
panel). Note that most ground-based stations selected are located in polluted areas, where big emission
sources of CO are present. We form the time-series from the ground-based stations by averaging spatially
over all the sites. We form the NR time-series similarly, but interpolate the NR surface data to the station
location. We do not add random observation errors to the NR time-series.

From Fig. 1, we see that, generally, the NR captures reasonably well the features of observed CO temporal
variability during the three phases characterizing the summer of 2003: before, during and after the heat wave
(the heat wave occurred on 31 July – 15 August). The correlation coefficient, $\rho$, between the ground-based
data and NR time-series shown in the middle panel is 0.71. From this, we conclude that the NR has a realistic
representation of the CO diurnal cycle. Note that CO concentration levels in the NR are slightly lower than
observed ones. The bias of the NR with respect to observed CO concentrations fluctuates around -10 % on
average during normal conditions and reaches -20% within the heat wave period. This means that the NR
reproduces the surface concentrations with a negative bias (NR lower than ground-based stations) between
10 and 20%. Nonetheless, the simulated CO concentrations and those measured by the ground-based stations
generally fall within the same range of values (between 200 and 400 $\mu gm^{-3}$). Thus, for the OSSE period
considered, we conclude that the NR is representative of the variability of actual observations over the
European domain, albeit with a negative bias.

Additionally, from Fig. 1 the behaviour of the CO time-series from the CR compared to the NR, is similar to
the behaviour of the NR CO time-series compared to the Airbase data. This suggests that the configuration of
our model is reasonably realistic, and reduces the likelihood that the OSSE produces overoptimistic results.



## 2.2 The S-5P CO simulated measurements

The S-5P will deploy the TROPOspheric Monitoring Instrument (TROPOMI) jointly developed by The
Netherlands and ESA (Veefkind et al. 2012). The TROPOMI instrument has heritage from both the OMI and
the SCIAMACHY missions. The TROPOMI instrument will make measurements in the UV-visible
wavelength range (270-500 nm), the near infrared, NIR (675-775 nm) and the shortwave infrared, SWIR
(2305-2385 nm). It will deliver a key set of gas and aerosol data products for air quality and climate
applications, including ozone, $NO_2$, formaldehyde, $SO_2$, methane and CO.

To enable sounding of the lower atmosphere at finer scales, TROPOMI has an unprecedented spatial
resolution of 7x7 $km^2$ at nadir. This relatively high spatial resolution is necessary for air quality applications
at local to regional scales. It will resolve emission sources with relatively high accuracy, and will obtain an
acceptable fraction of cloud-free spectra. In contrast to the advantages provided by the relatively high spatial
resolution of S-5P and design improvements, the SCIAMACHY CO data needs averaging in time (roughly
one month) and space (5x5 degrees) to obtain realistic CO distributions at comparable uncertainty (Galli et
al., 2012). Furthermore, TROPOMI will have a wide swath of 2600 km to allow for daily global coverage.
The relatively high radiometric sensitivity of S-5P will allow measurements at low albedo, thus helping track
smaller pollution events and improving the accuracy of air quality assessments and forecasts. The use of S-
5P CO total column measurements with inverse modelling techniques will also help quantify more accurately
biomass burning emissions and map their spatial distribution. The simultaneous measurements of CO and,
e.g. $NO_2$, will provide additional information on wildfire and other pollution episodes (Veefkind et al., 2012).

The NR results were used to generate a set of synthetic S-5P observations. This involves several steps: 1)
Generating realistic S-5P orbits and geolocation and viewing/solar geometries for the appropriate overpass
time; 2) Using the ECMWF modelled cloud distributions to generate effective cloud fractions; 3) Generation
of lookup tables for the averaging kernels and observation errors; 4) Collocation and application of the NR to
derive a set of synthetic observations for 3 Summer months and 3 Winter months. These steps are discussed
in the sub-sections below.





### 2.2.1 Orbit simulator


The System Tool Kit (STK, available from AGI, http://www.agi.com/products/) is used to generate the S-5P
orbit geometry and the geolocation of the edges of the swath as a function of time. Based on these
characteristics, the location of the individual observations with a spatial distance of 7 km are generated. Time
and longitude shifts are applied to the STK orbits to generate the orbits for the three Summer and three
Winter months. Subsequently, the solar and viewing geometries are computed. Finally, segments of the orbits
are maintained that have an overlap with the modelling domain.

### 2.2.2 Cloud properties


Cloud fields are obtained from the high-resolution operational weather forecast archive of the ECMWF.
Meteorological fields of liquid water content, ice water content, specific humidity and cloud fraction are
retrieved at a resolution of 0.25 x 0.25 degree for June-August 2003 and November 2003 - January 2004.
These quantities are converted to cloud optical properties. The optical properties determine the reflectance,
and are used to estimate effective cloud fractions and effective cloud top heights as would be retrieved from
the satellite observations (Acarreta et al., 2004). The distribution of effective cloud fractions was compared
with the distribution of effective cloud fractions obtained from OMI observations, and a reasonable
agreement was found for Summer and Winter months.

These effective cloud fractions (and corresponding cloud radiance fractions) are used to provide weights to
the cloud-free and cloud-covered fractions of the surface scene. The cloud altitude is used for the
computation of the averaging kernel.

### 2.2.3 Averaging kernel and measurement uncertainty lookup tables


Because of the large number of observations that will become available from the S-5P instrument, full
radiative transfer calculations for each observation separately is not feasible. We have chosen to build look-
up tables for a set of geometries based on a radiative transfer code that employs the adding-doubling method
in combination with optimal estimation (radiative transfer toolbox DISAMAR; de Haan, 2012). Look-up
tables are set up for the averaging kernels (1D vectors as a function of altitude) and the measurement





uncertainty. Results are stored for a number of surface albedos, cloud/surface pressures, solar zenith angles,
viewing zenith angles and relative azimuth angles. The look-up table details are provided in Table 1. Kernels
are provided on 21 pressure levels between 1050.0 and 0.1 hPa. Uncertainties are specified for clear-sky and
cloudy-sky separately.

Each simulation with DISAMAR consist of a forward calculation of the satellite-observed spectrum,
followed by a retrieval step based on the optimal estimation method (Rodgers, 2000). Instrument noise, listed
in Table 1, is converted into uncertainties for the retrieved CO column. A-priori trace gas profiles are taken
from the CAMELOT study (Levelt et al., 2009). We assume that both the cloud and the surface are
Lambertian reflectors. Kujanpää et al. (2015) provide further details of this procedure.

In particular the albedo is of major influence for the uncertainty, because it directly determines the signal
observed by the instrument. This dependence is shown in Fig.2. Over land, albedo values are typically of the
order of 0.1-0.2, with typical column errors of the order of 2 DU, or about $10^{17}$ molecules cm$^{-2}$. Because
typical CO columns over Europe are 2 $10^{18}$ molecules cm$^{-2}$, this is a relatively small error of the order of 5%.
These numbers are in good agreement with the results presented in the CO ATBD of TROPOMI (document
expected to be publicly released by the end of 2015). Over Sea, the albedo is very low, and the noise
dominates the signal. In order to simulate this behaviour in a realistic way we have added the albedo values
0.005, 0.01 and 0.02 to the albedo list.

We note that the uncertainties reported here are substantially lower than those reported for SCIAMACHY
(e.g. Gloudemans et al., 2008). This reflects a difference in specifications of the instruments, and the fact that
SCIAMACHY observations were hindered by ice build-up on the detectors. Real TROPOMI observations
will show if the relatively small errors are realistic.

***2.2.4 Synthetic observations generation***
The generation of the synthetic observations consists of the following steps:

• Collocation of the Nature run vertical profiles of CO to the locations of the observations.





• Computation of the effective cloud fraction, cloud radiance fraction, and cloud pressure from the
ECMWF cloud fields collocated to the observations.

• Collocation of the NIR albedo map (Surface albedo at 2300 nm is interpolated from a climatology
provided by SRON and based on SCIAMACHY observations (P. Tol, personal communication)) to
the locations of the observations.

• Extract interpolated values for the observation kernel and uncertainties from the look-up table.
• Compute the synthetic observation from the inner product of the kernel with the nature run CO
profile. This is done for both a clear sky and fully clouded situation, using the cloud pressure.

• Add a random noise amount to each observation, by drawing numbers from a Gaussian distribution
with a width determined from the uncertainty estimate.

• Compute the partially clouded synthetic observation by weighting the clear and cloudy results with
the cloud radiance fraction.


Over land, and in clear sky cases, the averaging kernel is close to 1, showing that the S-5P instrument is
observing the vertical column to a good approximation (see Fig. 3). In cloud-covered cases the kernel equals
0 for layers below the cloud pressure (yellow line in Fig. 3). For low-albedo cases (over ocean), Rayleigh
scattering becomes non-negligible, and the kernel is decreasing towards the surface, but the noise is
dominant in this case.

The results of this process is shown in Fig. 4. The figure demonstrates the high resolution of the NR (about 7
km) and the corresponding simulated amount of detail. The bottom panel shows the corresponding CO
observations. Over land the NR features are clearly present due to the relatively low uncertainty. Over the
ocean and Mediterranean the signal is dominated by the noise. An improved information content is observed
near Iceland, related to a thick cloud cover, where the higher signal reduces the relative noise.

**2.3 Pre-processing of S-5P CO total column observations**
This section describes the pre-processing of S-5P CO total column observations prior to assimilation into the
MOCAGE model (Peuch et al., 1999) for the OSSE simulations. Using the MOCAGE model for the AR and



CR simulations avoids the identical twin problem associated with using the same model for both the NR and
the OSSE simulations, which typically produces overoptimistic results (Arnold and Dey, 1986; Stoffelen et
al., 2006). Section 2.4 provides further details of the MOCAGE model.

The S-5P will produce large amounts of data owing to its wide swath and relatively high spatial resolution of
about 7x7 km$^2$. Thus, a pre-processing step is necessary to reduce the data volume for the data assimilation
experiments. For this study, we consider only pixels inside the OSSE simulation domain (Note that retrieval
pixels in each single cross-track are essentially instantaneous measurements of CO.). This has the advantage
of alleviating the data volume burden. However, a single cross-track over Europe could have more than
80,000 valid retrieval pixels. Furthermore, each individual pixel is associated with an averaging kernel vector
given at 34 vertical pressure levels, from the surface up to the top of the atmosphere (identified as 0.1 hPa).

Figure 3 shows an example of averaging kernels at the surface, as well as the averaging kernels
representative of retrievals including pixels with different cloud fractions (less than 10%, greater than 30%,
and greater than 80%). In addition, we discard data points with solar zenith angles larger than 80% or errors
exceeding 20%. The retrieval over sea is noise-dominated. Because of this, we only consider CO partial
columns above cloudy sea scenes with cloud fraction more than 80% and cloud top heights between the
surface and 650 hPa. Finally, we apply a spatially weighted mean to bin the measurements into 0.2° x 0.2°
grid boxes (~20 x 15 km at 45N), the assimilation model resolution; this is the set-up used for the OSSE
assimilation experiments (CR and AR), and is described in El Amraoui et al. (2008a). It combines the
MOCAGE model and the PALM (Projet d'Assimilation par Logiciel Multiméthode) data assimilation
module. Sections 2.4-2.5 provide further details of the CR and AR set-ups.

The weighted mean for pixels falling in the same model grid box is:

$$\tilde{c} = \frac{\sum_i w_i \, c_i}{\sum_i w_i}$$







375   where $\bar{c}$ is the weighted average, $c_i$ a single column measurement, and $w_i$ ($=1/\sigma_i^2$) is the inverse of the

376   variance corresponding to measurement $c_i$, and is the weight assigned to this single measurement. The

377   inverse of the variance associated with the weighted average is


$$\frac{1}{\sigma^2} = \sum_i w_i$$



381   The spatial binning not only reduces considerably the data volume but also results in an improved spatial

382   representativeness of the CO measurements by reducing the random error of each data pixel.


384   **2.4 The Control Run**

385   To generate the CR, it is important to use a state-of-the-art modelling system, which simulates the

386   observational data representing, for example, a current operational observational system. An important

387   requirement for an effective OSSE is to generate the CR with a model different to the one used to construct

388   the NR to avoid the identical twin problem (see Sect. 2.3). If the model from which we extract hypothetical

389   observations is the same as the assimilating model, the OSSE results tend to show unrealistic observation

390   impact and overly optimistic forecast skill (Arnold and Dey, 1986; Stoffelen et al., 2006). Consequently, by

391   using two independent models the OSSE will more realistically simulate the assimilation of real

392   observations. This follows our guiding principle to design an OSSE that is not too overoptimistic.

394   As mentioned in Sect. 2.3, we use the MOCAGE model to generate the CR. In this OSSE study, the CR is a

395   free model run. The MOCAGE model is a three-dimensional CTM developed at Météo France (Peuch et al.,

396   1999) providing the evolution of the atmospheric composition in accordance with dynamical, physical and

397   chemical processes. It provides a number of configurations with different domains and grid resolutions, as

398   well as various chemical and physical parameterization packages. Current use of MOCAGE includes several

399   applications: e.g., the Météo-France operational chemical weather forecasts (Dufour et al., 2004); the

400   Monitoring Atmospheric Composition and Climate (MACC) services (http://www.gmes-atmosphere.eu;

401   Márecal et al., 2015); and studies of climate trends of atmospheric composition (Teyssèdre et al., 2007).



Validation of MOCAGE simulations against a large number of measurements took place during the
Intercontinental Transport of Ozone and Precursors (ICARTT/ITOP) campaign (Bousserez et al., 2007).

In this study, we use a two-way nesting configuration to generate the CR and the AR (we describe the AR
set-up in Sect. 2.5): a global grid with a horizontal resolution of 2x2 degrees and a regional grid (5W-35E,
35N-70N) with a horizontal resolution of 0.2x0.2 degrees. The MOCAGE model includes 47 sigma-hybrid
vertical levels from the surface up to 5 hPa. The vertical resolution is 40 to 400 m in the boundary layer (7
levels) and about 800 m in the neighbourhood of the tropopause and in the lower stratosphere. The chemical
scheme used is RACMOBUS, which combines the stratospheric scheme REPROBUS (Lefèvre et al., 1994)
and the tropospheric scheme RACM (Stockwell et al., 1997). The RACMOBUS scheme includes 119
individual species, of which 89 are prognostic variables, and considers 372 chemical reactions.

We force the CR (and the AR) every 3 hours with the ARPEGE analysis (Courtier et al., 1991). We prescribe
the surface anthropogenic emission using the MACC-I emission database (https://gmes-
atmosphere.eu/about/project_structure/input_data/d_emis/). We do not include the fire emissions in the
CR and AR experiments described in this paper, as they are a priori not known. This means that any
signature of fire emissions in the AR (see Sect. 2.5) can only come from assimilation of the CO
measurements. Note that for the NR, the surface anthropogenic emissions come from the MACC-II
inventory, helping to differentiate the CR from the NR. As for the NR, the CR has a spin-up period of three
months.

**2.5 The Assimilation run**
We assimilate simulated S-5P total column CO observations derived from the LOTOS-EUROS NR into the
MOCAGE CTM at a 0.2° spatial resolution using the MACC extended domain (5W-35E, 35N-70N). The
assimilation system used in this study is MOCAGE-PALM (e.g., El Amraoui et al., 2008a) developed jointly
by Météo-France and CERFACS (Centre Européen de Recherche et de Formation Avancée en Calcul
Scientifique) in the framework of the ASSET European project (Lahoz et al., 2007b). The assimilation
module used in this study is PALM, a modular and flexible software, which consists of elementary
components that exchange data (Lagarde et al., 2001). It manages the dynamic launching of the coupled





components (forecast model, algebra operators and input/output of observational data) and the parallel data
exchanges. Massart et al. (2009) used the assimilation system MOCAGE-PALM to assess the quality of
satellite ozone measurements. The MOCAGE-PALM assimilation system also helps identify and overcome
model deficiencies. In this context, its assimilation product has been used in many atmospheric studies in
relation to ozone loss in the Arctic vortex (El Amraoui et al., 2008a); tropics/mid-latitudes exchange
(Bencherif et al., 2007); stratosphere-troposphere exchange (Semane et al., 2007); and exchange between the
polar vortex and mid-latitudes (El Amraoui et al., 2008b). For this OSSE, to speed up the assimilation
process we use the 3D-Var version of PALM. In the OSSE, the MOCAGE model provides the CR and by
assimilating the simulated CO data from the NR, the MOCAGE model provides the AR. Thus, we produce
the CR and AR outputs with a model different from that used to produce the NR (see Sect. 2.1).

A key element of the data assimilation system is the background error covariance matrix (the **B**-matrix)
(Bannister, 2008). It has a large impact on the 3D-Var analysis used in this study and, thus, it is important to
use a form of **B** that is as realistic as possible. In MOCAGE-PALM, we base the **B**-matrix formulation on the
diffusion equation approach (Weaver and Courtier, 2001). It can be fully specified by means of the 3-D
standard deviation field (square root of the diagonal elements of **B**, in concentration units or as a percentage
of the background field) and 3-D fields of the horizontal ($L_x$ and $L_y$) and vertical ($L_z$) local correlation
length-scales. We can estimate the **B**-matrix elements more efficiently using an ensemble method (Bannister,
2008). This technique consists of feeding an ensemble of states through the data assimilation system to
simulate the important sources of error. However, this approach is time-consuming and, therefore, not used in
this study.

For this study, we use a simple parameterization for the **B**-matrix: $L_x$ and $L_y$ are assumed homogeneous and
equal to 35 km (about two model grid lengths); $L_z$ is constant and set to one vertical model layer. As in Emili
et al. (2014), the background standard deviation 3-D field is parameterized as a vertically varying percentage
of the background profile, which decreases from values of 25% at the surface to values of 15% in the upper
troposphere, and decreases further throughout the stratosphere to values of 5% in the upper stratosphere (not
shown). We base these settings on several 1-day assimilation trials; they ensure reasonable values of standard
self-consistency tests, e.g., providing chi-squared ($\chi^2$) values close to 1 (see Fig. 3 in Sect. 3.1). Furthermore,





a value of $L_x$ and $L_y$ of 35 km corresponds to more than one grid length of the model, allowing the model to
resolve these features. The data assimilation procedure will weight both the observations and the model 1-
hour forecasts (from the last analysis point), and will update locations not coincident with the observations
through the correlation length-scales. Table 2 summarizes the parameters used for the assimilation
experiments.
**3. Results**
**3.1 Evaluation of the assimilation run**
In this section, we evaluate the impact of the assimilation of the S-5P CO total column. First, we evaluate the
consistency of the assimilation run by separating the clear-sky pixels from their cloudy counterparts (Sect.
3.1.1). Second, to further understand the impact on the surface CO field of the simulated S-5P CO total
column measurements, we investigate the analysis increment ($\delta x$) to provide a quantitative diagnostic of the
quality of the analysis for a selected date, 15 June 2003 (Sect. 3.1.2).

*3.1.1 Consistency of the assimilation run*
We have performed two OSSEs. The first one includes all pixels in the OSSE domain, regardless of whether
they are cloudy or clear-sky and the second only includes clear-sky pixels. A pixel is considered as clear
when the cloud fraction is less than 10%. Comparison of the ARs from these two OSSEs indicated that the
impact of including all pixels is small. The largest differences between the respective ARs in relation to the
NR are 4% in regions over North Europe (North Sea and Scandinavia), with the AR for clear-sky pixels
closer to the NR (not shown). We can explain these results by the fact the summer generally has low amounts
of cloud. Consequently, we only present the results from the OSSE with all pixels.
To evaluate the AR, we calculate the $\chi^2$ diagnostic associated with the Observation minus Forecast (OmF)
differences (see, e.g., Lahoz et al., 2007a). Here, we normalize the OmF differences by the background error.
We also calculate histograms of the Observation minus Analysis (OmA) differences, the observation and the
simulation from the CR (observation-minus-control run, hereafter OmC) differences, and the OmF





differences. We use the observational error to normalize the differences building the histograms of OmA,
OmC and OmF.

Figure 5 (top panel) shows the chi-squared time-series for OmF and its associated auto-correlation function
calculated over the three-month period of the OSSE experiments, computed as daily averages. The chi-
squared diagnostic starts with a maximum of about 1.56, and takes values down to 0.75, with a mean of 0.9
over the OSSE three-month period. The chi-squared time-series is nearly stable since it exhibits relatively
small variability (a standard deviation of about 0.14). Furthermore, the auto-correlation of the chi-squared
statistic drops to zero, with no correlation after a time delay of 20 days. The calculation of the auto-
correlation shows that the chi-squared statistic is uncorrelated after a time lag of 20 days; this means that
after this time the mathematical expectation $E(\chi^2)$ is equal to the average of the chi-squared statistics. We
find $E(\chi^2) = 0.90$, which is close to the theoretical value of 1 (see Lahoz et al., 2007a). This result indicates
that the a priori error statistics as represented in the **B**-matrix slightly overestimate the actual error statistics
from the OmF differences.

To test whether the observations, forecast and analysis fields, and their associated errors, are consistent with
each other, we calculate the histograms of OmA, OmF and OmC only over land (normalized by the
observation error) over the three-month period (Fig. 5, bottom panel). For a properly set up assimilation
system, the OmF and OmA normalized histograms should be close to a Gaussian distribution with mean zero
and standard deviation one. Figure 3 (bottom panel) shows that the OmA and OmF differences are close to a
Gaussian distribution centred near to or at zero. The OmF has a mean and standard deviation of 0.10 and
1.73, respectively, whereas the OmA has nearly a zero mean and a standard deviation of 1.05. This indicates
that the centre of the OmA histogram is closer to zero and more peaked than the histogram of OmF. We
expect this, since the analyses should be closer to the observations than the forecasts. Furthermore, the
histogram for OmA indicates that the errors in the **R**-matrix, the observational counterpart of the **B**-matrix,
are a good representation of the analysis error.

Based on the above results, we conclude that the background error covariance matrix, **B**, and its
observational counterpart, **R**, prescribed in our assimilation system are reasonably well characterized (see,





e.g., Lahoz et al., 2007a, for a discussion of the specification of errors in a data assimilation system).
Furthermore, the above results are consistent with the assumption that the errors in the observations and the
forecasts are Gaussian.

The shape of the OmC normalized histogram, which has a mean and standard deviation of 2.36 and 5.60,
respectively, indicates the presence of a relatively large bias between the S-5P observations and the CR. The
assimilation reduces this bias, as shown by the analyses being significantly closer to the observations than
the simulation from the CR. This shows that the assimilation of simulated S-5P CO total column
observations has a significant impact on the CO forecasts and analyses.

***3.1.2 Study of increments***
To understand further the impact on the surface CO field of the simulated S-5P CO total column
measurements, we calculate the analysis increment ($\delta x$) for a single analysis time at 14:00 UTC on 15 June
2003. We calculate this increment as the analysis minus the model first guess (1-hour forecast). The analysis
increment provides a quantitative diagnostic of the quality of the analysis (see, e.g., Fitzmaurice and Bras,

2008).


Figure 6 (top panel) shows the spatial distribution of $\delta x$ at the model surface. One can see the spread of the
impact of the simulated observations across large regions. This is owing to S-5P having a wide swath
allowing it to sample larger regions. The most substantial corrections are over land, where there are sufficient
observations to have an impact.  Over sea, the increments tend to be negligible, as any observations found
there have relatively large errors. Thus, there will not be much difference between the model first guess and
the analysis. Likewise, this is also true in the regions outside the satellite footprint.

To provide further insight into the impact of S-5P CO measurements, we calculate latitude-height and
longitude-height cross-sections at 48.8N, 2.6E, near Paris, for 15 June 2003. Figure 4 (bottom left and
bottom right panels) shows a zoom of the zonal and meridional vertical slices of the analysis increment. We
see significant corrections to the model first guess (identified by large increments) confined to a deep layer.
These corrections are larger at the surface, and exhibit a second maximum around 650 hPa. This vertical





structure is mainly attributable to the forecast error standard deviation (given as a vertically varying fraction
of the local CO mixing ratio), the square root of the diagonal entry of the **B**-matrix, and which is higher in
the boundary layer (where the value of the S-5P CO averaging kernel is close to 1). The shape of the S-5P
analysis increments also exhibits a second peak around 650 hPa. The increments for this particular day thus
show a clear impact from the S-5P CO measurements in the PBL and the free troposphere.

The shape of the S-5P increments is similar to that of typical SCIAMACHY analysis increments, which also
extend through a deep layer and have a maximum at the surface (Tangborn et al., 2009). The fact that both
these analysis increments stretch out over a deep layer is owing to similarities in the S-5P and SCIAMACHY
averaging kernels - both are close to unity over cloud-free land (see Fig. 4). Note that the situation shown in
Fig. 6 is a snapshot and depends on the particular conditions for this time. An average of the increments over
the summer period would tend to show a uniform distribution in height.

**3.2 Evaluation of the summer OSSE**
*3.2.1 Summer averages*
Figure 7 shows the fields of surface CO from the CR, and the NR and the AR, averaged over the northern
summer period. One can see the general change of CO over land between the CR (top left panel) and the AR
(bottom panel). We can ascribe this to the contribution of simulated S-5P total column CO data sampled from
the NR. This figure shows several differences between the CR and AR fields that indicate the superior
behaviour of the AR in capturing features in the NR. For example, over Eastern Europe and Russia, the AR
CO concentration values are closer to those in the NR; in particular, the CR shows generally lower values
than in the NR. Nevertheless, over Portugal, where the NR shows the forest fires that occurred over the
summer, the AR captures them only slightly better than the CR. We expect the relatively poor performance of
the CR regarding fires, as the fires are not included in the CR set-up (see Sect. 2.4). Although the AR
captures the forest fires slightly better than the CR (through assimilation of CO measurements), the relatively
poor temporal resolution of the S-5P ultimately limits its performance. A geostationary satellite, given its
relatively high temporal resolution, should be able to capture better the temporal variability of these forest
fires (Edwards et al., 2009).



***3.2.2 Statistical metrics***
In this section, we provide a quantitative assessment of the benefit from S-5P CO total column measurements
on the CO surface analysis. For this, we perform a statistical analysis of the different OSSE experiments for
northern summer 2003.

We calculate the mean bias (MB, in parts per billion by volume, ppbv), its magnitude reduction (MBMR,
ppbv), and the root mean square error (RMSE, ppbv), and its reduction rate (RMSERR, %). Note that
although recent papers have raised concerns over the use of the RMSE metric (Willmott and Matsuura, 2005;
Willmott et al., 2009), Chai and Draxler (2014) discuss circumstances where the RMSE is more beneficial.
We use the correlation coefficient, ρ to measure the linear dependence between two datasets, and the fraction
of the true variability (i.e., variability represented by the NR) reproduced by the CR or AR.

For a single model grid box, we define the statistical metrics (MB, RMSE, ρ) with respect to the NR as:

$$MB(X) = \frac{1}{N}\sum(X - NR)$$

$$MBMR = |MB(CR)| - |MB(AR)|$$

$$RMSE(X) = \sqrt{\frac{1}{N}\sum(X - NR)^2}$$

$$RMSERR = 100 \times \left(1 - \frac{RMSE(AR)}{RMSE(CR)}\right)$$

$$\rho(X) = \frac{\sum(X - \overline{X})(NR - \overline{NR})}{\sqrt{\sum(X - \overline{X})^2 \sum(NR - \overline{NR})^2}}$$




where X denotes the CR or the AR; N is the number of data samples; the vertical bars denote the absolute
value operator; and the overbar symbol represents the arithmetic mean operator. The MB metric gives the
average value by which the CR or the AR differs from the NR over the entire dataset.

*3.2.3 Results of the statistical tests*
Figure 8 presents the zonal and meridional means of the difference between the CR and the AR averaged
over the northern summer 2003 (1 June – 31 August). We also plot the confidence interval representing the
areas where the AR is not significantly different to the CR at the 99% confidence limit (highlighted in the
grey colour). These two figures show that there is benefit from the S-5P CO total column data over the first
few bottom levels of the troposphere, i.e., the lowermost troposphere. Between the surface and 800 hPa, a
negative peak is present in the zonal difference field (over Scandinavia), and in the meridional difference
field (over Eastern Europe). Note that the zonal field shows two areas, one with positive values and the other
with negative values representing a CR greater than the AR and a CR smaller than the AR, respectively. The
positive peak, at a slightly higher level (i.e., lower pressure) than the negative peak, is representative of the
Mediterranean Sea, whereas the negative peak is more representative of the land areas (Scandinavia and
Eastern Europe). Figure 8 indicates that the S-5P CO corrects the model in the lower troposphere with a
larger impact over land and with a less large impact in the PBL. This is consistent with the behaviour of the
analysis increments shown in Fig. 6.

Figure 9 shows the performance of the biases between the CR and the NR, and the AR and the NR at the
surface, and averaged over the northern summer of 2003 (1 June – 31 August). The MBMR, which compares
the magnitude of the CR vs NR and AR vs NR biases, indicates the geographical areas where the simulated
S5P CO total column data have the most impact. The MBMR shows that the AR is closer to the NR than the
CR, almost everywhere in the domain (reflected by the prevalence of the red colours in the bottom left
panel). This indicates that the simulated S-5P CO total column data generally provide a benefit at the surface,
and especially over land areas where the CO sources are sparse.

We also calculate the RMSE as well as the reduction rate of the RMSE, RMSERR (Figure 10), both keeping
the systematic error (Fig. 10 top), and removing the systematic error (Fig. 10 bottom). We calculate the





systematic error in the AR and CR by subtracting the NR field from each of them, producing a debiased AR
and CR. For the case where we remove the systematic error, we perform the statistics on the debiased AR
and CR. If we examine the RMSE statistics, Fig. 8 shows that the CR gets closer to the NR over the Atlantic
Ocean and over the Eastern domain including Russia and Scandinavia, when we remove the systematic error.
For example, over these areas we obtain ~30 ppbv and ~10 ppbv for the RMSE keeping and removing the
systematic error, respectively. For the reduction of the RMSE, RMSERR, the behaviour for the CR is similar
overall, showing a reduction rate of 60% and 30-45% keeping and removing the systematic error,
respectively. Note that over Scandinavia the reduction rate goes down from 60% to about 10% after
removing the systematic error.

These results indicate that S-5P CO data show more benefit when keeping the systematic error in the
calculation of the RMSE. Following our guiding principle of avoiding an overoptimistic OSSE, we consider
only the values of RMSE obtained when we remove the systematic error. For this case, the average reduction
rate for the AR is around 20-25% over land (except Scandinavia) and close to 10% over sea and over
Scandinavia.

In Figure 11, we show the correlation between the CR and the NR, and the correlation between the AR and
the NR, at the surface for the three northern summer months (1 June – 31 August). Figure 11 shows that the
AR is closer than the CR to the NR with the correlation coefficient reaching 0.9 over land. By contrast, the
correlation coefficient between the CR and the NR is typically less than 0.5, with very low values over
Eastern Europe, where CO sources are sparse.

*3.2.4 Time-series at selected locations*
Figure 12 shows time-series from the NR, the CR and the AR over the three areas of the study domain
represented by the squares shown in Figs. 9 (bottom panel) and 10 (right panels). (i) The Paris region (Fig.
12, top panel); (ii) a region over Portugal, where forest fires occurred during the northern summer (Fig. 12,
middle panel); and (iii) an area in the Eastern part of the study domain, where the reduction of RMSE (i.e.,
RMSERR) was much larger than for other regions (Fig. 12, bottom panel). For all three areas, the AR is
generally closer to the NR than the CR, showing the impact of the simulated observations. We calculate the




biases between the AR and CR vs the NR by computing the difference NR-X, where X is AR or CR, and
normalizing by the number of observations over the northern summer period. The biases are: (i) Paris region,
CR: 48 ppbv, AR: 38 ppbv; (ii) Portugal, CR: 101 ppbv, AR: 83 ppbv; (iii) Eastern part of domain: CR: 21
ppbv, AR: 5 ppbv.

Over Paris (top panel), the CR is already close to the NR and the impact of the S-5P CO simulated
observations is small. Over Portugal (middle panel), the presence of fires is not seen in the CR (e.g., a
maximum of CO at the beginning of the heat wave), as the fires were not taken into account in the CR as
they are not known a priori (see Sect. 2.4). In contrast, over this specific location we see the impact of the
fires on the CO concentrations in the AR with, however, much lower values than for the NR. During the
fires, the CO concentrations in the AR over Portugal were larger than 500 ppbv, whereas the CR remained
relatively unchanged with concentrations less than 200 ppbv. Over the Eastern part of the study area (bottom
panel), the temporal variability is not high and the magnitude of the bias between the CR and the NR is
small, but it is removed in the AR.

*3.2.5 Sensitivity tests for fire episode*
The assimilation system we use has a default criterion to discard CO column observations with values larger
than 75% of the MOCAGE value. This criterion is not appropriate to situations resulting in excessive values
in the CO concentrations, as is the case for forest fires. To understand further the performance of the OSSE
over the period of the Portugal forest fires we perform a second OSSE without this default criterion. This
second OSSE covers the period of the forest fires (25 July – 15 August). For this second OSSE, we compare
the total column values and the surface values of the CO fields from the CR and the AR (Figs. 13-15,
respectively).

Figure 13 shows the CO total column at 14:15 UTC on 4 August 2003 (during the period of the Portugal
forest fires) from the NR (top left panel); the simulated S-5P observations (top right panel); the CR (bottom
left panel); and the AR (bottom right panel). We can see that the AR captures the fire event, indicated by
relatively high values of the CO total column over Portugal, whereas the CR does not. This confirms the
results shown in Fig. 12, which highlight the benefit provided by the S-5P CO total column measurements, in





particular regarding the capture of the signature of the Portugal forest fires. Note that the S-5P measurement
is noise-dominated over the sea (top right panel). This accounts for the sharp edge in the CO total column
field seen between the Iberian Peninsula and the Bay of Biscay for the AR (bottom right panel).

Figure 14 shows the time-series of the surface CO concentrations over the period 25 July – 15 August (that
of the Portugal forest fires). In comparison to the original OSSE (see middle panel of Fig. 12), the AR is now
closer to the NR, having now peak values of about 900 ppbv, instead of peak values of about 550 ppbv. The
CR still has peak values less than 200 ppbv. This indicates that the relatively low values in the AR (in
comparison to the NR) for the original OSSE shown in the middle panel of Fig. 12 result from the
application of the default criterion to discard CO column observations that are far away from MOCAGE
values. The results from Fig. 14 confirm those shown in Fig. 13, and reinforce the benefit provided by the S-
5P CO total column measurements, in particular regarding the capture of the signature of the Portugal forest
fires.

## 4. Conclusions

We perform a regional-scale Observing System Simulation Experiment (OSSE) over Europe to explore the
impact of the LEO satellite mission S-5P carbon monoxide (CO) total column measurements on lowermost
tropospheric air pollution analyses, with a focus on CO surface concentrations and the Planetary Boundary
Layer (PBL). The PBL varies in depth throughout the year, but is contained within the lowermost
troposphere (heights 0-3 km), and typically spans the heights 0-1 km. We focus on northern summer 2003,
which experienced a severe heat wave with severe societal impact.

This OSSE study provides insight on the impact from LEO S-5P CO measurements on surface CO
information. We perform the standard steps of an OSSE for air quality. (i) Production of a Nature Run, NR.
(ii) Test of the realism of the NR. (iii) Different models to produce, on the one hand, the NR, and on the other
hand, the OSSE experiments to create the Control Run, CR, and the Assimilation Run, AR. (iv) Calculation
of synthetic observations, observation uncertainty, and averaging kernels to represent sensitivity of the
observations in the vertical. (v) Quantitative evaluation of the OSSE results, including performing statistical



significance tests, and self-consistency and chi-squared tests. Based on the specifications of the TROPOMI
instrument, relatively low CO column uncertainties of around 5% are anticipated over the European
continent.

Our guiding principle in the set-up of this OSSE study is to avoid overoptimistic results. To achieve this, we
address several factors considered likely to contribute to an overoptimistic OSSE. (i) We use different
models for the NR and the OSSE experiments. (ii) We check that the differences between the NR and actual
measurements of CO are comparable to the CO field differences between the model used for the OSSE and
the NR. (iii) We remove the systematic error (calculated as the bias against the NR) in the OSSE outputs (AR
and CR) and compare the debiased results to the NR.

The OSSE results indicate that simulated S-5P CO total column measurements during northern summer 2003
benefit efforts to monitor surface CO. The largest benefit occurs over land in remote regions (Eastern
Europe, including Russia) where CO sources are sparse. Over these land areas, and for the case when we
remove the systematic error, we obtain a lower RMSE value (by ~10 ppbv) for the AR than for the CR, in
both cases vs the NR. Over sea and Scandinavia, we also obtain a lower RMSE (by ~10%) for the AR than
for the CR, in both cases vs the NR. Consistent with this behaviour, we find the AR is generally closer to the
NR than the CR to the NR, with a correlation coefficient reaching 0.9 over land (NR vs AR). By contrast, the
correlation coefficient between the CR and the NR is typically less than 0.5, with very low values over
Eastern Europe, where CO sources are sparse. In general, for all the metrics calculated in this paper, there is
an overall benefit over land from the S-5P CO total column measurements. Significance tests on the CR and
AR results indicate that, generally, the differences in their performance are significant at the 99% confidence
level. This indicates that the S-5P CO total column measurements provide a significant benefit to monitor
surface CO.

We further show that, locally, the AR is capable of reproducing the peak in the CO distribution at the surface
due to forest fires (albeit, weaker than the NR signal), even if the CR does not have the signature of the fires
in its emission inventory. A second OSSE shows that this relatively weak signal of the forest fires in the AR
arises from the use of a default criterion to discard CO total column observations too far from model values,





a criterion not appropriate to situations resulting in excessive values in the CO concentrations, as is the case
for forest fires. This second OSSE shows a much stronger signal in the AR, which is now much closer to the
NR than the CR, confirming the benefit of S-5P CO total column measurements.

Further work will involve extending the OSSE approach to other S-5P measurements, such as ozone total
column, and $NO_2$ and formaldehyde tropospheric columns. These studies will complement similar studies on
the benefit from Sentinel-4 and -5 measurements. Collectively, these OSSE studies will provide insight into
the relative benefits from the Sentinel-4, -5 and -5P platforms for monitoring atmospheric pollution
processes.

**5. Acknowledgments**
Support for this work came partly from the ESA funded project "Impact of Spaceborne Observations on
Tropospheric Composition Analysis and Forecast" (ISOTROP –ESA contract number 4000105743). WAL
acknowledges support from an internal project from NILU. RA, JLA, PR, LE and WAL acknowledge
support from the RTRA/STAE. JK and JT acknowledge support from the Academy of Finland (Project no.

267442).




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





**Tables**

**Table 1:** Spectral and radiometric settings for DISAMAR, and the look-up table nodes.

| Spectral and radiometric settings | |
| --- | --- |
| Spectral range [nm] | 2330-2345 |
| Spectral resolution (FWHM) [nm] | 0.25 |
| Spectral sampling [nm] | 0.1 |
| SNR Earth radiance | 120 |
| SNR Solar irradiance | 5000 |
| Additional calibration error (%) | 1.0, correlation length 100 nm |
| **Node points** | |
| cos(SZA) | 0.1 - 1.0, step 0.1 |
| cos(VZA) | 0.3 - 1.0, step 0.1 |
| Relative azimuth [degree] | 0.0, 180.0 |
| Cloud/surface pressure | 1100 - 200, step -100 |
| Cloud/surface albedo | 0.0, 0.005, 0.01, 0.02, 0.04, 0.06, 0.1, 0.2, 0.3, 0.4, 0.8, 0.9 |
| Pressure layers | 1100, 1000, 900, 800, 700, 600, 500, 400, 300, 200, 137.50, 68.75, 34.38, 17.19, 8.59, 4.30, 2.15, 1.07, 0.54, 0.27, 0.13, 0.07 |










**Table 2**: Description of the configuration used in the assimilation system

|  | **Description** |
|---|---|
| Assimilation | 3D-var, 1 hour window |
| Background standard deviation | in % of the background field (vertically variable) |
| Background correlation zonal Length scale ($L_x$) | constant 35 km |
| Background correlation meridional length scale ($L_y$) | constant 35 km |
| Background correlation vertical length scale ($L_z$) | one vertical model layer |
| S-5P total column CO observation errors | from retrieval product and weighting to account for the total column |
















**Figures**

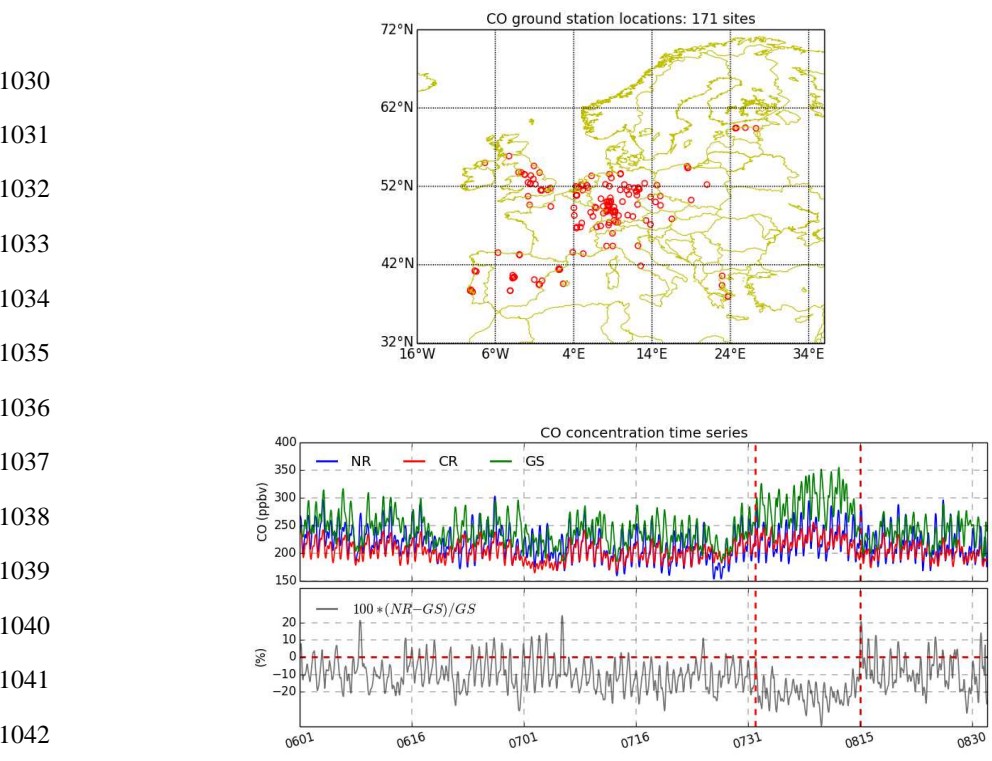

**Figure 1:** Top panel: location of selected ground-based stations for CO measurements taken from the Airbase
database during northern summer 2003 (1 June – 31 August). There are 171 sites with locations shown by
circles. The labels show longitude, degrees (x-axis) by latitude, degrees (y-axis). Middle panel: simulated
and measured time-series of CO concentrations in surface air from nature run (blue line), the control run (red
line) and from the selected 171 Airbase sites (green line). We form the CO time-series for the ground-based
stations by averaging the hourly data representative of the 171 sites. The labels show time in MMDD format
(x-axis) by CO concentration, parts per billion by volume, ppbv (y-axis). Bottom panel: The gray curve
shows the relative error of the nature run with respect to the observations, defined as NR value minus ground
station value divided by the ground station value and multiplied by 100. The labels show time in MMDD
format (x-axis) by relative error, percent (y-axis). The vertical red dashed lines in the middle and bottom
panels delineate the 2003 European heat wave period (31 July – 15 August).






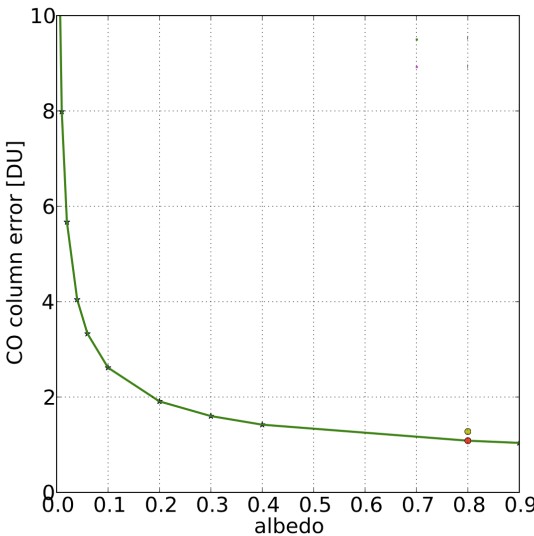

**Figure 2:** Dependence of the CO column uncertainty on the surface albedo. Simulation settings are: solar zenith angle 53 degrees, viewing zenith angle 26 degrees, relative azimuth angle 0 degree, cloud/surface pressure 1100 hPa.














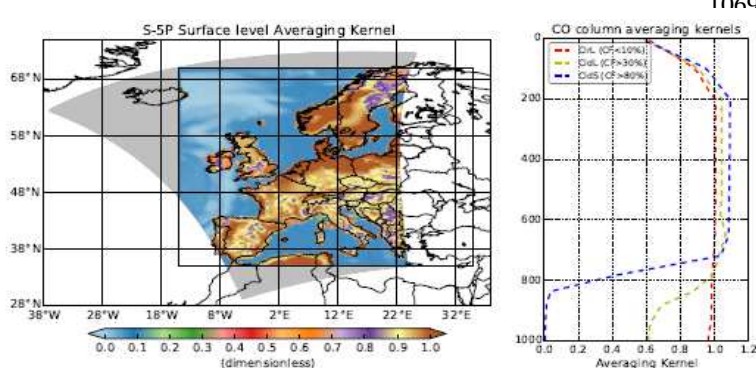

1077

**Figure 3:** Left panel: S-5P CO averaging kernel values at the surface. Labels are longitude, degrees (x-axis)

by latitude, degrees (y-axis). Right panel: Averaging kernels for land pixels with cloud fraction less than 10%

(dashed red lines); for land pixels with cloud fraction greater than 30% (dashed yellow lines); and for sea

pixels with cloud fraction greater than 80% (dashed blue lines). The averaging kernels are for an average of

the data shown on the swath for 1 June 2003 at 12:34 UTC. Labels are averaging kernel, normalized (x-axis)

by pressure level, hPa (y-axis).

















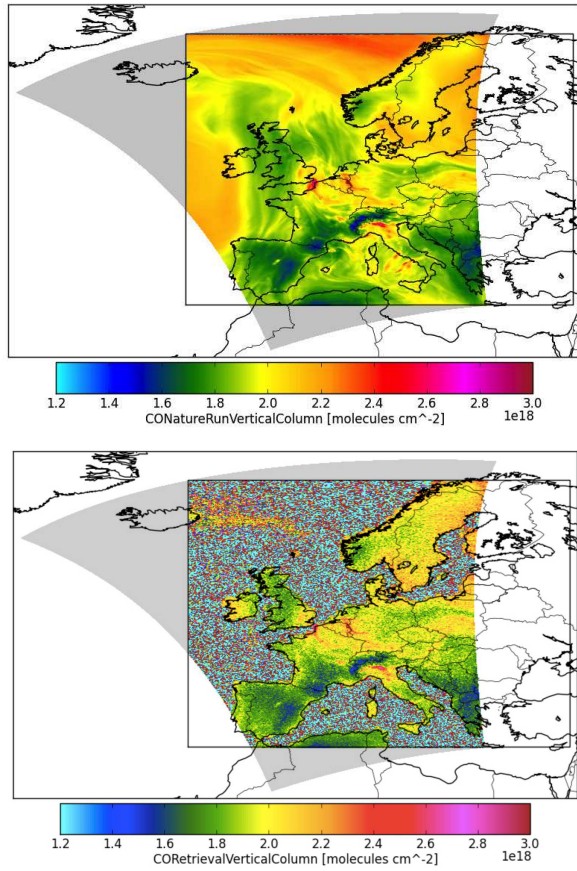

**Figure 4**: Top: Nature run collocated to the synthetic S-5P observations for the 12:34 orbit on 1 June 2003.

Bottom: corresponding synthetic observations.
























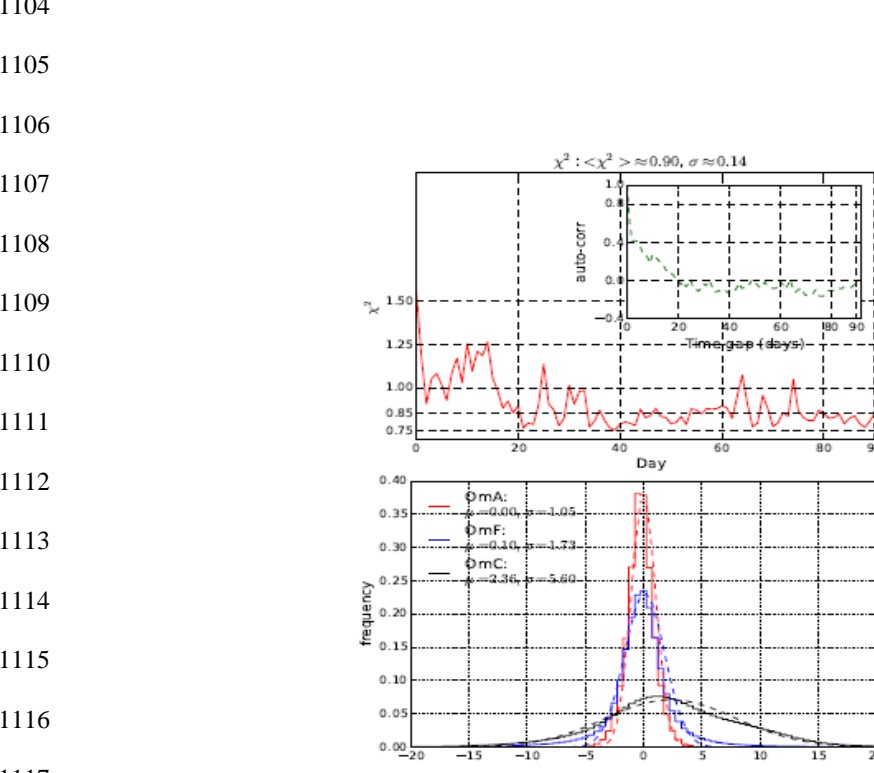

**Figure 5:** Self-consistency tests. Top panel: time-series (red line) of $\chi^2$ for OmF and its associated auto-
correlation signal (green line). For the $\chi^2$ diagnostic we normalize the OmF differences by the background
error. The labels show time, days (x-axis) and $\chi^2$ value (y-axis) for the $\chi^2$ plot, and time gap, days (x-axis) and
auto-correlation (y-axis) for the auto-correlation plot. Bottom panel: histograms of Observations minus
Analysis (OmA -red solid line), Observations minus Forecast (OmF -blue solid line), and Observations
minus Control run (OmC -black solid line). We normalize these differences by the observation error. The
dashed lines correspond to the Gaussian fits of the different histograms. The labels show the OmA, OmF or
OmC differences (x-axis) and the frequency of occurrence of the differences (y-axis). We calculate the
diagnostics OmA, OmF, and OmC over the period of 1 June – 31 August 2003.








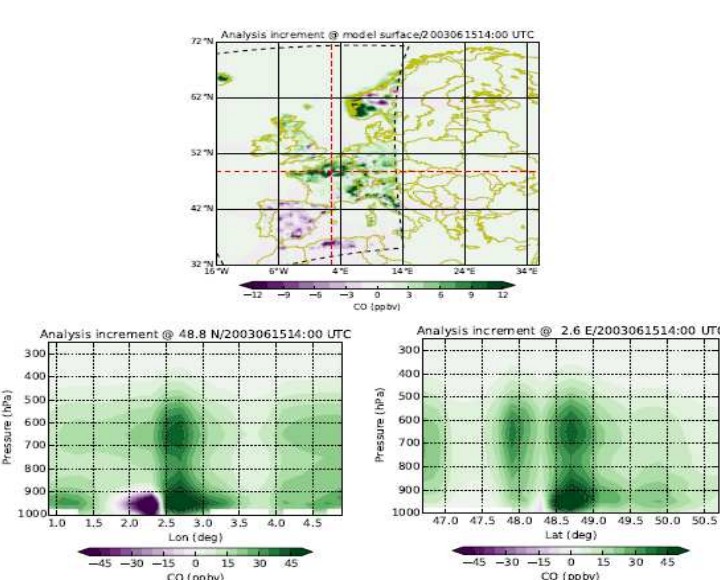

**Figure 6**: S-5P CO analysis increments, units of ppbv, at 14:00 UTC on 15 June 2003: Top panel:
geographical distribution at the model surface. Red dashed lines show zonal and meridional vertical slices at
48°8 N, and 2°6 E, respectively. The black dashed line shows the S-5P cross-track at 13:12 UTC, clipped to
fit the OSSE simulation domain. Note that we measure the S-5P CO observations at 13:12 UTC. The labels
show longitude, degrees (x-axis) and latitude, degrees (y-axis). Left and right bottom panels show,
respectively, the longitude-height and latitude-height cross-sections at a location near Paris. The labels for
the bottom panels show longitude, degrees (x-axis, left panel), latitude, degrees (x-axis, right panel), and
pressure, hPa (y-axis, both panels). Green/purple colours indicate positive/negative values in the increment
fields.








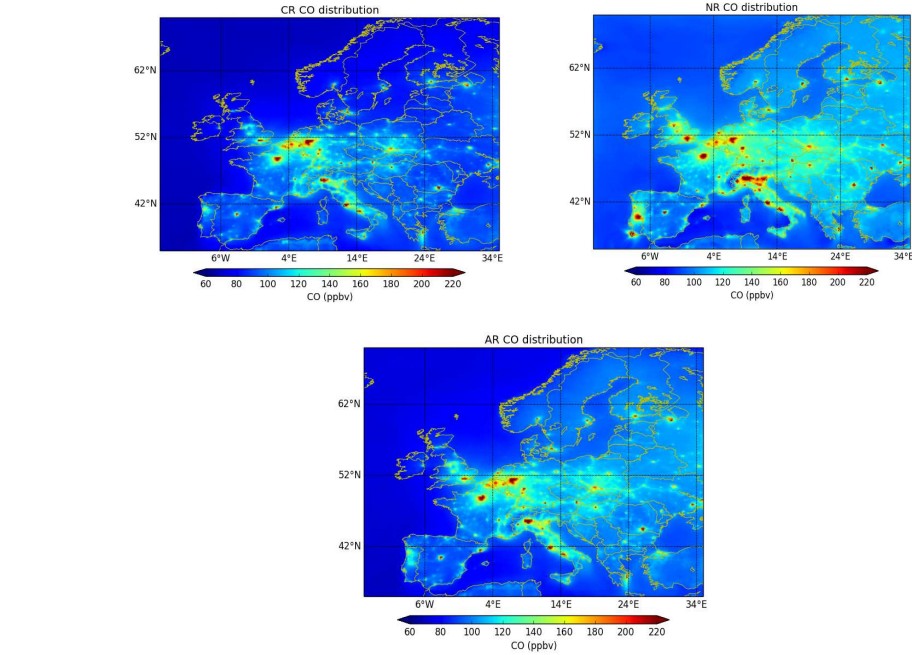


**Figure 7:** Distribution of CO surface concentrations, units ppbv, averaged for the period 1 June – 31 August
2003. Top left panel: the control run (CR) from MOCAGE; right top panel: the nature run (NR) from
LOTOS-EUROS; bottom panel: the assimilation run (AR) from MOCAGE obtained after assimilating the S-
5P CO total column simulated data sampled from the NR. In all panels, the labels show longitude, degrees
(x-axis) and latitude, degrees (y-axis). Red/blue colours indicate relatively high/low values of the CO surface
concentrations.













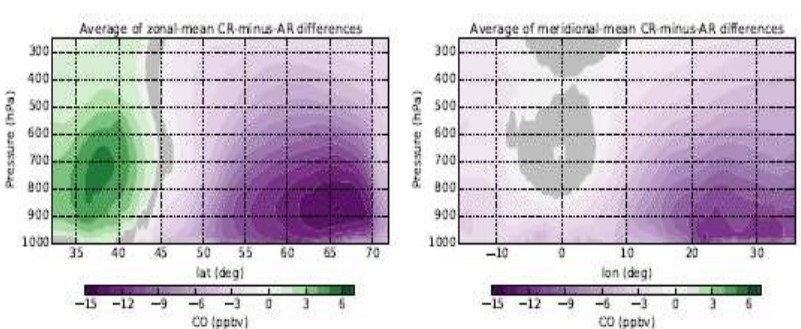

1183

**Figure 8:** Zonal (left panel) and meridional (right panel) slices of the difference between the CR and AR CO

fields, units of ppbv, averaged over the summer period (1 June – 31 August 2003). The areas highlighted in

grey colour indicate where the AR is not significantly different to the CR at the 99% confidence level. The

labels in the left panel are latitude, degrees (x-axis) and pressure, hPa (y-axis). The labels in the right panel

are longitude, degrees (x-axis) and pressure, hPa (y-axis). Green/purple colours indicate positive/negative

values in the difference fields.


















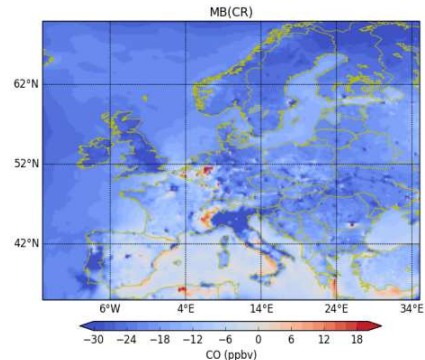 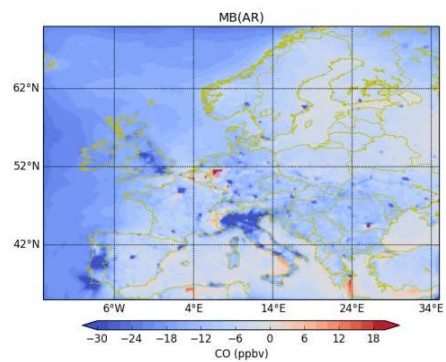

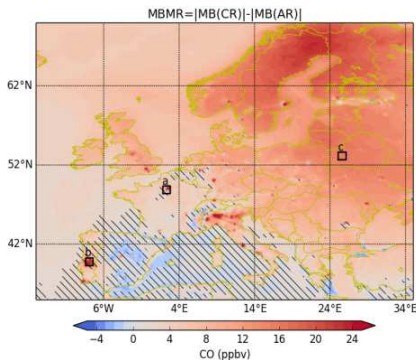


**Figure 9:** Mean bias reduction at the surface for CO, units of ppbv: Left top panel shows the CR mean bias
with respect to the NR (CR-NR). Right top panel shows the AR mean bias with respect to the NR (AR-NR).
Bottom panel shows the mean bias magnitude reduction (absolute value of the mean bias for CR minus the
absolute value of the mean bias for AR). We average the data over northern summer 2003 (1 June – 31
August). The labels show longitude, degrees (x-axis) and latitude, degrees (y-axis). The hatched area in the
bottom panel shows where the mean bias plotted in this panel (MBMR) is not statistically significant at the
99% confidence level. The three squares in the bottom panel represent locations for the three time-series
shown in Fig. 12. Red/blue colours indicate positive/negative values in the MB/MBMR.








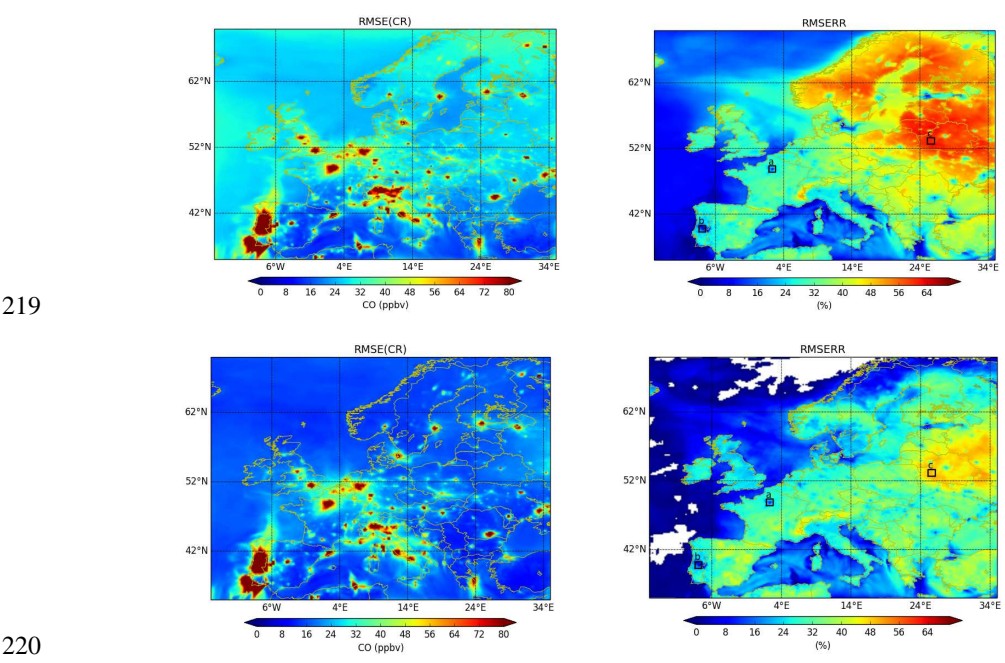




**Figure 10:** Top: Root Mean Square Error (RMSE), units of ppbv, between CR and NR (left panel), and its
corresponding reduction rate RMSERR, in % (right panel) keeping the systematic error. Bottom: Same as top
panel but calculating the RMSE after removing the systematic error. The labels on each panel are longitude,
degrees (x-axis) and latitude, degrees (y-axis). The three squares in the two right panels represent locations
for the three time-series shown in Fig. 12. Red/blue colours indicate relatively high/low values in the
RMSE/RMSERR.














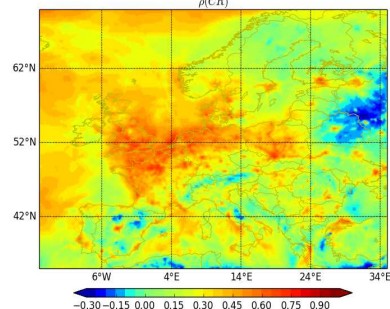
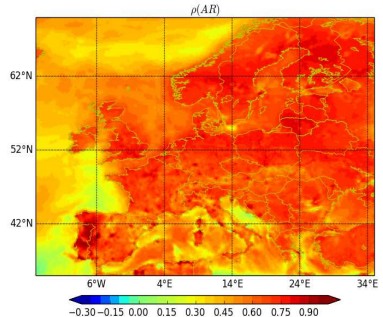


**Figure 11:** Correlation coefficient between the CR and the NR (left panel) and the AR and the NR (right panel) at the surface and for the northern summer period (1 June – 31 August). The labels are longitude, degrees (x-axis) and latitude, degrees (y-axis). Red/blue colours indicate positive/negative values of the correlation coefficient.





















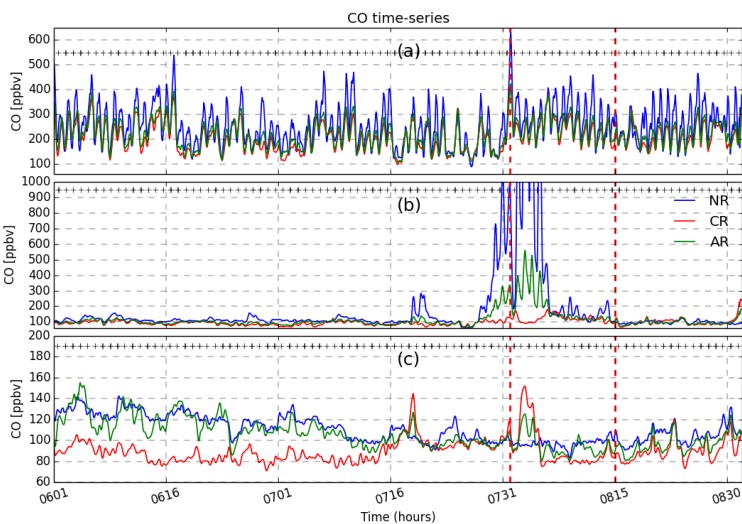


**Figure 12:** Time-series for CO surface concentrations (1 June – 31 August) from NR (blue colour), CR (red
colour) and AR (green colour) over three different locations represented by squares in Figs. 9 and 10. Top
panel: area near Paris; middle panel: area over Portugal, where forest fires occurred; bottom panel: Eastern
part of the study domain. The labels in the three panels are time, in format MMDD (x-axis) and CO
concentration, ppbv (y-axis). The plus symbols at the top of each panel indicate availability of observations
from the S-5P platform.















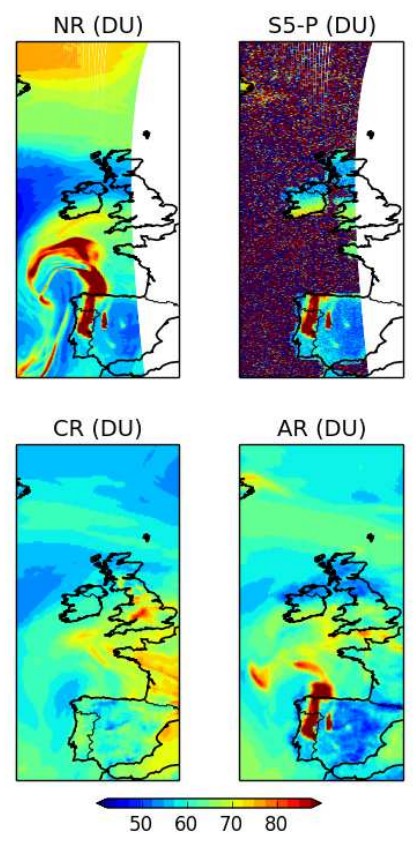


**Figure 13:** CO total column at 14:15 UTC on 4 August 2003, Dobson units, DU. Top left panel: NR; top
right panel: simulated S-5P observations; bottom left panel: CR; bottom right panel: AR. Red/blue colours
indicate relatively high/low values of the CO total column.














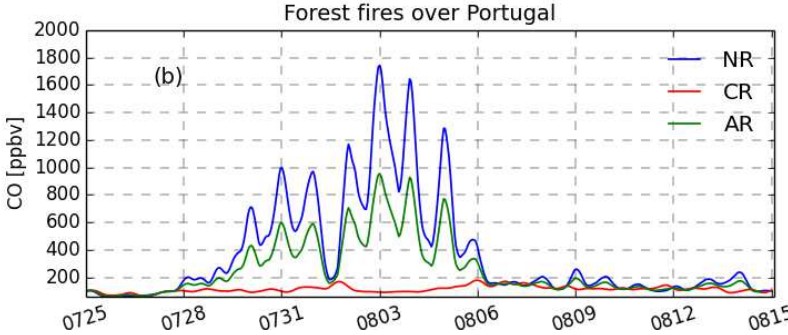



**Figure 14:** Time-series for CO surface concentrations for the period covering the Portugal forest fires (25
July – 15 August) from NR (blue colour), CR (red colour) and AR (green colour) over the location associated
with the middle panel of Fig.12. These data concern the second OSSE we perform to understand the
behaviour of the original OSSE over the period of the forest fires (see text for more details). The labels are
time, in format MMDD (x-axis) and CO concentration, ppbv (y-axis).






