# Peer review of "Impact of Spaceborne Carbon Monoxide Observations from the S-5P platform on"

_Atmospheric Chemistry and Physics, 2015_

## Referee Comment (RC1) · Anonymous Referee #1 · 24 Mar 2016

This manuscript describes an Observing System Simulation Experiment that quantifies the impact of the TROPOMI CO total column from a future Sentinel-5 Precursor platform on tropospheric analyses and forecast of the MOCAGE model. The authors use a set of different models (the LOTOS-EUROS and the TM5) to develop their Nature Run to avoid over-optimization in the OSSE system. They also carried out a specific study on the assimilation of fires using MOCAGE by modifying the model regular setup. This is a very good study from a team of experts in the field of data assimilation and OSSE, so I recommend publishing after addressing the minor issues below. The writing for sessions "2.4 The Control Run" and "2.5 The Assimilation Run" need care, more than what I suggest below. Also, make sure all acronyms are properly defined.

L30: "...with the largest benefit occurring over land in remote regions", explain remote from what? Sources?

L60-68: several sensors are listed but all references are from the MOPITT team. Please add references for the relevant sensors.

L81-103 paragraph: In the discussion of S-5, S-4, and S-5P, etc, please list the time frame of these missions. Since OMI and SCIAMACHY are discussed, as well as S-5P, this should be a good place to introduce TROPOMI. Among the sensors/missions discussed in this paragraph, which ones have CO, since it's the topic of study here?

L99-101: "The S-5P LEO platform will address the challenge of limited revisit time from LEOs by providing unprecedented high spatial resolution of 7x7 km, and improved sensitivity in the Planetary Boundary Layer (PBL), allowing resolution of, e.g., derived CO emission sources at finer scales than hitherto." How?

L223: "...the NR has a realistic representation of the CO diurnal cycle." Does CO have diurnal cycle? Also, describe the ground measurement methods. Is it in situ or radiometric?

L232-233: "...the behaviour of the CO time-series from the CR compared to the NR, is similar to the behaviour of the NR CO time-series compared to the Airbase data." Not clear, are the differences similar? You might want to add the difference 100*(NR-CR)/NR.

L237-242: TROPOMI should have been introduced in the Introduction.

L308: "Over sea, ..." should probably be "Over the ocean, ..."

L387: "...different to ..." to "...different from ..."

L391: "... the OSSE will more realistically simulate..." to "... the OSSE will simulate more realistically ..."

L392: "This follows our guiding principle to ..." to "This allows us to ..."

L394-395: "As mentioned in Sect. 2.3, we use the MOCAGE model to generate the CR. In this OSSE study, the CR is a 395 free model run." to "In this OSSE study, the CR is a free model run using MOCAGE."

L409: ". . . and about 800 m in the neighbourhood of the tropopause . . ." to ". . . and approximately 800 m near the tropopause . . ."

L417: . . . as they are a priori not known." To ". . . as their a priori is not known."

L420: ". . ., helping to differentiate the CR from the NR." To ". . ., which helps to differentiate the CR from the NR."

L420: "As for the NR, . . ." to "Similar to the NR, . . ."

L435-436: add "the"

L453-454: ". . . for the B-matrix: Lx and Ly are . . .; Lz is constant and . . ." to ". . .for the B-matrix, where Lx and Ly are . . .; and Lz is constant and . . ."

L459: ". . . (see Fig. 3 in Sect. 3.1)." should be Fig. 5

L504: Should be Figure 5, not Figure 3

L564-570: The reasons the AR are not performing well over fire emissions are not explained correctly in this paragraph, as suggested by L670-676. Should move part of L670-676 to this part of the paper to properly explain why AR did not work well over fires.

L642: "Figure 11 shows that the AR . . ." would be better to "The AR . . ."

L651: ". . . (iii) an area in the Easter part of the study domain, . . ." should name the location/country/regions.

L662-665: same issue explaining MOCAGE assimilation over fires.

Fig. 5. Legends and labels are not legible when overlap with dashed lines. Should redo with care and use larger fonts.

Fig. 7 and alike: when use multiple panels, try to arrange them to maximize the space and show larger figures. Please use larger fonts for titles.

Fig. 13, the top right panel title should be S-5P, not S5-P

---

## Referee Comment (RC2) · Anonymous Referee #2 · 10 May 2016

This manuscript describes an Observing System Simulation Experiment to assess the impact of Sentinel 5p CO observations on CO concentrations over Europe. Their guiding principle–a laudable one–is avoid the common critique that OSSE results are overly optimistic. They have partially succeeded in that approach. They have gone through the effort of choosing two different nested chemical transport models, one for their nature run (NR) and the other for their assimilation (AR) and control run (CR). They also have a reasonable retrieval model simulator to construct observational operators for S5p. They sample the NR using an S5P simulator and then assimilate those pseudo retrievals into their AR and CR. They have been careful to compare their NR to independent observations to assess it realism. However, this is where I have some

concerns. While they establish a bias and variance between the NR and independent observations, their fundamental threshold is that the difference appears "reasonable". What they have not done is to relate the errors between the NR and independent observations to the interpretation of the performance of the AR. So, if the accuracy and precision of the NR is twice as bad, what should one infer about the performance of the AR? I would argue that a more important implementation of their guiding principle is to assimilate real observations, e.g., MOPITT and/or SCIAMACHY, into their system and compare the analysis fields to independent observations. Then, they could do an OSSE for the same observing system and assess the statistical difference between the AR and NR sampled at independent observations versus AR(real) against independent observations. That would provide a better sense of what the OSSE limitations actually are. As it stands, I'm still suspicious of the overall performance. Furthermore, we don't know how well S5p will perform given other sensors, e.g., MOPITT, CrIS, are already taking CO data with comparable performance.

Otherwise, the overall work is reasonable and the authors have performed some nice statistical analysis of the results. Of course, in principle, this OSSE should have been performed *before* S5p was funded to assess its potential. But, practice is still catching up with theory. I've attached comments of the manuscript in the accompanying pdf.

Please also note the supplement to this comment:
http://www.atmos-chem-phys-discuss.net/acp-2015-924/acp-2015-924-RC2-supplement.pdf

**Supplement:**

[revised manuscript text omitted]

**4-1** May 9, 2016, 6:40 PM, Ref
Need to include CrIS.

**4-2** May 9, 2016, 6:40 PM, Ref
What about planned?

**4-3** May 9, 2016, 6:40 PM, Ref
Why is methane defined twice?

**4-4** May 9, 2016, 6:40 PM, Ref
How is this point relevant if you're not discussing geostationary options?

**5-1** May 9, 2016, 6:40 PM, Ref
Relative to what, MOPITT?  Not clear how since they both have NIR channels.

**5-2** May 9, 2016, 6:40 PM, Ref
Poorly formed sentence.  And not quite true.  The resolution of an inverse emission estimate is controlled by the data and transport/diffusion.  Not clear which is the limiting factor without analysis.

**5-3** May 9, 2016, 6:40 PM, Ref
True, but how is relevant?

**5-4** May 9, 2016, 6:40 PM, Ref
So, TROPOMI needs to be assessed relative to the existing satellites.

**6-1** May 9, 2016, 6:40 PM, Ref
So, if one knew CO concentrations or emissions better, what societal or scientific benefit would have been achieved?  Improved forecasts?  Better attribution?

**7-1** May 9, 2016, 6:40 PM, Ref
Shouldn't the control run include assimilation of the existing observing system, e.g., MOPITT, AIRS, CrIS, rather than a free run?  This would be true for your 2003 test case when MOPITT and AIRS was available.

**7-2** May 9, 2016, 6:40 PM, Ref
I shouldn't have to look up the figure.  How does it relate to the specific OSSE elements listed?

**9-1** May 9, 2016, 6:40 PM, Ref
Did I miss something?  Where is the CR described?

**9-2** May 9, 2016, 6:40 PM, Ref
There is also some high frequency component that is missed. What is that frequency?  is that the night time values?  Needs to be discussed.

**9-3** May 9, 2016, 6:40 PM, Ref
That statement needs to be limited to the ultimate performance of the OSSE.  The assimilation can't saying anything better than 10-20% in accuracy.

**10-1** May 9, 2016, 6:40 PM, Ref
At what accuracy?

**10-2** May 9, 2016, 6:40 PM, Ref
How low?

**Notes**

**10-3**
May 9, 2016, 6:40 PM, Ref
More accurately compared to what?

**11-1**
May 10, 2016, 9:54 AM, Ref
But OMI is quite a bit larger in footprint than TROPOMI. MODIS would be a better choice. Describe how the cloud fraction is related to the 7km footprint. How different are cloud ODS between the UV and the NIR? It seems like you are assuming they are the same.

**12-1**
May 9, 2016, 6:40 PM, Ref
provides

**12-2**
May 9, 2016, 6:40 PM, Ref
Is it released now?

**12-3**
May 9, 2016, 6:40 PM, Ref
sea

**13-1**
May 10, 2016, 9:54 AM, Ref
Cloudy

**13-2**
May 10, 2016, 9:54 AM, Ref
It looks like you are assuming that the retrievals will work under partially cloudy scenes. MOPITT NIR only works under clear sky. Provide a reference on NIR CO retrievals under partially cloudy conditions and justify why a weighted approach works. Also I haven't heard any discussion of aerosols. These will be important for emissions like biomass burning and industry.

**13-3**
May 10, 2016, 9:54 AM, Ref
This is the first time MOCAGE is mentioned even though it is implicitly referenced in the CR run. Needs to be mentioned earlier.

**16-1**
May 10, 2016, 9:54 AM, Ref
That is not exactly true. The existence of burning and the burnt area can be obtained from optical measurements. That is very important a priori information. Is that being ignored?

**17-1**
May 10, 2016, 9:54 AM, Ref
In the introduction, the authors argued for the value of TROPOMI to resolve emissions. Here the focus has shifted to concentration estimation. What is the scientific rationale? What are the limitations of this OSSE, then, to make statements about resolving sources?

**21-1**
May 10, 2016, 9:54 AM, Ref
The correction in the free trop points to the role of boundary conditions in the assimilation, which would be an important consideration for a GEO. How much change is occurring at the boundary of the nested grid?

**21-2**
May 10, 2016, 9:54 AM, Ref
That is not obvious. The biggest weakness is using 3D-var and having poor prior statistics. Given that significant fires generally last longer than a day, a proper inversion system would pick those up. That does not diminish the value of GEO sounders.

**21-3**
May 10, 2016, 9:54 AM, Ref
Please elaborate as to whether it is the diurnal sampling or the effective sampling density that is more important for a geo.

**Notes**

**21-4** May 10, 2016, 9:54 AM, Ref

That's too qualitative. Could please provide some simple metrics, e.g., means, to quantify these statements?

**23-1** May 10, 2016, 9:54 AM, Ref

Why?

**24-1** May 10, 2016, 9:54 AM, Ref unbiased

**24-2** May 10, 2016, 9:54 AM, Ref

Why is this called a systematic error? It looks like you are merely removing the bias term in the error, which is simply a statement that the assimilation is not an unbiased estimator.

**25-1** May 10, 2016, 9:54 AM, Ref

However, both the CR and AR miss the high frequency min/max. Why?

**25-2** May 10, 2016, 9:54 AM, Ref

This is a weakness of the OSSE design, not the measurements.

**25-3** May 10, 2016, 9:54 AM, Ref

Please explain why the variability is high in Paris but not in E. Europe.

**25-4** May 10, 2016, 9:54 AM, Ref

This seems like a discovery for the authors post-facto. I recommend using this version of the OSSE only rather than devoting a whole section to it. It's what should have been done originally.

**26-1** May 10, 2016, 9:54 AM, Ref

The first several paragraphs are repetitious of the introduction. I recommend removing. In fact, the OSSE has shown that S5P will have a similar or bigger impact on the free trop rather than merely the surface.

---

## Author Comment (AC1) · 30 Jun 2016

Referee # 1

We thank the referee for his/her helpful comments. Our response to referee #1 is below. Items in bold and italics are the referee comments.

**L30: ". . .with the largest benefit occurring over land in remote regions", explain remote from what? Sources?**

We mean regions far away from important CO sources. We will rephrase this in the revised version.

**L60-68: several sensors are listed but all references are from the MOPITT team. Please add references for the relevant sensors.**

We add additionnal references in particular for AIRS and IASI CO in the revised version.

**L81-103 paragraph: In the discussion of S-5, S-4, and S-5P, etc, please list the timeframe of these missions. Since OMI and SCIAMACHY are discussed, as well as S-5P, this should be a good place to introduce TROPOMI. Among the sensors/missions discussed in this paragraph, which ones have CO, since it's the topic of study here?**

- We will mention the time frame for S-4. S-5 and S-5P time frames are already mentioned (see lines 92-94).

- We will mention TROPOMI in this paragraph but note we describe the instrument characteristics and show how the the S-5P simulated measurements are generated in section 2.2.

- We will replace GOME-2 by IASI which measures CO and has the same revisit local time.

**L99-101: "The S-5P LEO platform will address the challenge of limited revisit time from LEOs by providing unprecedented high spatial resolution of 7x7 km, and improved sensitivity in the Planetary Boundary Layer (PBL), allowing resolution of, e.g., derived CO emission sources at finer scales than hitherto." How?**

We will clarify this point in the revised version by adding information on source inversion.

**L223: ". . .the NR has a realistic representation of the CO diurnal cycle." Does COhave diurnal cycle? Also, describe the ground measurement methods. Is it in situ or radiometric?**

- At air surface usually carbon monoxide exhibits a diurnal variation, generally with two peaks, one in the morning and the second in the evening.

- Ground station measurements are real in situ observations taken from AirBase data set. We will mention this in the revised version

**L232-233: ". . .the behaviour of the CO time-series from the CR compared to the NR,is similar to the behaviour of the NR CO time-series compared to the Airbase data." Not clear, are the differences similar? You might want to add the difference 100*(NRCR)/NR.**

We likely are in the configuration where the NR is between the GS data and the CR with similar behaviour that means in our assimilation we may need a similar correction for the CR to obtain the NR or for the NR to obtain the GS. We will rephrase this in the revised version.

**L237-242: TROPOMI should have been introduced in the Introduction.**

See previous answer.

**L308: "Over sea, . . ." should probably be "Over the ocean, . . ."**

*We will correct this.*

**L387: ". . .different to . . ." to ". . .different from . . ."**

*We will correct this.*

**L391: ". . . the OSSE will more realistically simulate. . ." to ". . . the OSSE will simulate more realistically . . ."**

*we will correct this.*

**L392: "This follows our guiding principle to . . ." to "This allows us to . . ."**

*we will correct this.*

**L394-395: "As mentioned in Sect. 2.3, we use the MOCAGE model to generate the CR. In this OSSE study, the CR is a free model run." to "In this OSSE study, the CR is a free model run using MOCAGE."**

*we will correct this.*

**L409: ". . . and about 800 m in the neighbourhood of the tropopause . . ." to ". . . and approximately 800 m near the tropopause . . ."**

*we will correct this.*

**L417: . . . as they are a priori not known." To ". . . as their a priori is not known."**

*We will correct this.*

**L420: ". . ., helping to differentiate the CR from the NR." To ". . ., which helps to differentiate the CR from the NR."**

*we will correct this.*

**L420: "As for the NR, . . ." to "Similar to the NR, . . ."**

*We will correct this*

**L435-436: add "the"**

we will correct this.

**L453-454: ". . . for the B-matrix: Lx and Ly are . . .; Lz is constant and . . ." to ". . .for the B-matrix, where Lx and Ly are . . .; and Lz is constant and . . ."**

we will correct this.

**L459: ". . . (see Fig. 3 in Sect. 3.1)." should be Fig. 5**

*We will correct this.*

**L504: Should be Figure 5, not Figure 3**

We will correct this.

**L564-570: The reasons the AR are not performing well over fire emissions are not explained correctly in this paragraph, as suggested by L670-676. Should move part of L670-676 to this part of the paper to properly explain why AR did not work well over fires.**

We will take into account this suggestion.

**L642: "Figure 11 shows that the AR . . ." would be better to "The AR . . ."**

We will correct this in the revised version.

**L651: ". . . (iii) an area in the Easter part of the study domain, . . ." should name the location/country/regions.**

*We will correct this in the revised version.*

**L662-665: same issue explaining MOCAGE assimilation over fires.**

*We will rephrase this paragraph and give more details.*

**Fig. 5. Legends and labels are not legible when overlap with dashed lines. Should redo with care and use larger fonts.**

*We will improve the quality of this figure in the revised version.*

---

## Author Comment (AC2) · 30 Jun 2016

Referee # 2

We thank the referee for his/her helpful comments. Our response to referee #2 is below. Items in bold and italics are the referee comments.

*While they establish a bias and variance between the NR and independent observations, their fundamental threshold is that the difference appears "reasonable".*
*What they have not done is to relate the errors between the NR and independent observations to the interpretation of the performance of the AR. So, if the accuracy and precision of the NR is twice as bad, what should one infer about the performance of the AR?*

We will rewrite this part of the paper avoiding vague statements and quantifying the differences between the NR and the GS and the CR and the GS. We will also compare these differences to typical errors in the MOCAGE data assimilation system derived from published papers (e.g., work by El Amraoui et al., 2014).

*I would argue that a more important implementation of their guiding principle is to assimilate real observations, e.g., MOPITT and/or SCIAMACHY, into their system and compare the analysis fields to independent observations. Then, they could do an OSSE for the same observing system and assess the statistical difference between the AR and NR sampled at independent observations versus AR(real) against independent observations. That would provide a better sense of what the OSSE limitations actually are.*

We think that the suggestion of the referee is valuable, and suitable for a study in its own right. However, it would only provide direct information on the error of the OSSE for MOPITT and/or SCIAMACHY CO and not directly for S5P. The S5P instrument has different characteristics to MOPITT and SCIAMACHY. Instead, and to keep the study tractable, we focus on the evaluation of the uncertainty in the NR, and compare it to published studies using the MOCAGE DA system. This in line with standard practice in AQ OSSEs as discussed in Timmermans et al (2015).

*As it stands, I'm still suspicious of the overall performance. Furthermore, we don't know how well S5p will perform given other sensors, e.g., MOPITT, CrIS, are already taking CO data with comparable performance.*

In this AQ OSSE, we follow standard practice by comparing the performance of S5P against a free model run. If the proposed satellite data are to have added value, they must perform better than a model. This is the first step one must take to evaluate the added value of a proposed satellite instrument. We will mention this in the revised text.

*Otherwise, the overall work is reasonable and the authors have performed some nice statistical analysis of the results. Of course, in principle, this OSSE should have been performed \*before\* S5p was funded to assess its potential. But, practice is still catching up with theory. I've attached comments of the manuscript in the accompanying pdf.*

We thank the referee for this comment. We address below the other points from the referee.

*4-1*
*Need to include CrIS.*

We will add information on CrIS

*4-2*
*What about planned?*

We will check if there are plans for GEO missions to measure CO.

*4-3*
**Why is methane defined twice?**

We identify that the formula for methane is $CH_4$.

*4-4*
**How is this point relevant if you're not discussing geostationary options?**

We think it helps the reader to contrast the characteristics of GEO and LEO satellite platforms with respect to atmospheric composition.

*5-1*
**Relative to what, MOPITT? Not clear how since they both have NIR channels.**

We mean that the S5P with its SWIR band will do better than our model in the PBL. Furthermore, compared to TIR instruments such as IASI, we expect S5P to do better in the PBL (Veefkind et al., 2012). We will mention this in the revised text.

*5-2*
**Poorly formed sentence. And not quite true. The resolution of an inverse emission estimate is controlled by the data and transport/diffusion. Not clear which is the limiting factor without analysis.**

We will reword the sentence and take account of the referee's comment.

*5-3*
**True, but how is relevant?**

We think it is helpful to remind the reader where the PBL is located.

*5-4*
**So, TROPOMI needs to be assessed relative to the existing satellites.**

See response to the general comments from this referee.

*6-1*
**So, if one knew CO concentrations or emissions better, what societal or scientific benefit would have been achieved? Improved forecasts? Better attribution?**

We will provide here an example of the benefits of improved knowledge of the CO distribution.

*7-1*
**Shouldn't the control run include assimilation of the existing observing system, e.g., MOPITT, AIRS, CrIS, rather than a free run? This would be true for your 2003 test case when MOPITT and AIRS was available.**

See the response to the general comments of this referee.

*7-2*
**I shouldn't have to look up the figure. How does it relate to the specific OSSE elements listed?**

We will provide this figure and relate it to the OSSE elements listed (this will be Fig. 1 in the revised paper). We will renumber the other figures.

*9-1*
***Did I miss something? Where is the CR described?***

We will refer to section 2.4, which describes the CR.

*9-2*
***There is also some high frequency component that is missed. What is that frequency? is that the night time values? Needs to be discussed.***

We will discuss this high frequency component.

*9-3*
***That statement needs to be limited to the ultimate performance of the OSSE. The assimilation can't saying anything better than 10-20% in accuracy.***

We will rephrase this sentence following the referee's suggestion.

*10-1*
***At what accuracy?***

We will quote Veefkind et al (2012) on this point: 15% (accuracy) and 10% (precision).

*10-2*
***How low?***

We will quote Veefkind et al on this point (the value is 2%)

*10-3*
***More accurately compared to what?***

We will rephrase the sentence.

Original sentence reads:

"The use of S-5P CO total column measurements with inverse modelling techniques will also help quantify more accurately biomass burning emissions and map their spatial distribution."

A simple solution would be: remove "more accurately"

The statement refers to the current observing system, consisting of, e.g., IASI, MOPITT, OMI, GOME-2, including measurements of the species CO and $NO_2$. S5P will provide global coverage, enhanced sensitivity for CO at the surface (compared to, e.g., IASI) and 3.5 to 7 km high spatial resolution observations.

*11-1*
***But OMI is quite a bit larger in footprint than TROPOMI. MODIS would be a better choice. Describe how the cloud fraction is related to the 7km footprint. How different are cloud ODS between the UV and the NIR? It seems like you are assuming they are the same.***

The cloud fractions were derived at the resolution of the ECMWF 0.25 x 0.25 degree grid. This is ca. 30 x 30 $km^2$ at the equator and decreases as a function of latitude. The ground pixel of OMI UV-2 and VIS channels is 13 x 24 $km^2$ at nadir increasing to 13 x 128 $km^2$ at edges of the swath. We consider that the ECMWF grid cells and OMI pixels are of comparable size for the purpose of comparing the cloud fraction distributions (ca. 0.5 million pixels or cells in each distribution). We model clouds with a simple Lambertian reflectors and ignore any wavelength dependency of cloud fraction.

We will include this information in the revised version.

*12.1*
*Provides*

We think provide is appropriate

*12-2*
*Is it released now?*

Yes, it is available from the ESA Sentinel-5P TROPOMI document library:

https://sentinel.esa.int/web/sentinel/user-guides/sentinel-5p-tropomi/document-library.

We will provide this information in the revised paper.

*12.3*
*sea*

We will correct this.

*13.1*
*Cloudy*

We understand clouded and cloudy are both appropriate here. If insisted upon, we will change this.

*13-2*
*It looks like you are assuming that the retrievals will work under partially cloudy scenes. MOPITT NIR only works under clear sky. Provide a reference on NIR CO retrievals under partially cloudy conditions and justify why a weighted approach works. Also I haven't heard any discussion of aerosols. These will be important for emissions like biomass burning and industry.*

TROPOMI NIR CO retrievals in partially cloudy conditions are discussed in Landgraf et al.: "Carbon monoxide total column retrievals from TROPOMI shortwave infrared measurements", Atmos. Meas. Tech. Discuss., doi:10.5194/amt-2016-114. They are also discussed in Vidot et al.:"Carbon monoxide from shortwave infrared reflectance measurements: A new retrieval approach for clear-sky and partially cloudy atmospheres", Remote Sens. Environ., doi:10.1016/j.rse.2011.09.032. We will add these references to the paper. We did not include aerosol effects in our study.

*13-3*
*This is the first time MOCAGE is mentioned even though it is implicitly referenced in the CR run. Needs to be mentioned earlier.*

At the start of section 2, when we first mention the CR, we will introduce MOCAGE and provide appropriate references.

*16-1*
*That is not exactly true. The existence of burning and the burnt area can be obtained from optical measurements. That is very important a priori information. Is that being ignored?*

The visible information on burnt area and burning does not provide knowledge on CO concentrations. We will add this information in the text.

*17-1*
*In the introduction, the authors argued for the value of TROPOMI to resolve emissions. Here the focus has shifted to concentration estimation. What is the scientific rationale? What are the limitations of this OSSE, then, to make statements about resolving sources?*

We will reword the text in the introduction to state that we focus on CO concentrations. We will clarify the scientific rationale of the paper and indicate the limitations of the OSSE.

*21-1*
*The correction in the free trop points to the role of boundary conditions in the assimilation, which would be an important consideration for a GEO. How much change is occurring at the boundary of the nested grid?*

First, note that the OSSE concerns S5-P, which is a LEO. Nevertheless, we focus on the surface level and we assume that the effect of the free troposphere on the boundary layer is secondary. Due to the revisit time of S5-P, we expect the impact at the boundaries to be small. Second, for efficiency reasons and storage limitations, we set up our DA system to only store the data over the regional domain. This means that without rerunning the OSSE, we cannot quantify the response to the reviewer's comment. If requested by the reviewer, we could rerun the OSSE for a short period and address the reviewer's question.

*21-2*
*That is not obvious. The biggest weakness is using 3D-var and having poor prior statistics. Given that significant fires generally last longer than a day, a proper inversion system would pick those up. That does not diminish the value of GEO sounders.*

We will clarify that we are talking about concentrations and not emission inversions.

*21-3*
*Please elaborate as to whether it is the diurnal sampling or the effective sampling density that is more important for a geo.*

The comment from the referee is not clear to us. We think that diurnal sampling (high temporal resolution) will be the determining factor, as the relatively coarse model resolution would compromise the high spatial sampling.

*21-4*
*That's too qualitative. Could please provide some simple metrics, e.g., means, to quantify these statements?*

We will quantify this difference in the revised paper.

*23-1*
*Why?*

Owing to the relatively small variability of CO over remote land regions, the S5-P data can provide a larger benefit compared to regions where the variability is relatively high. We will make this point in the revised version.

*24.124*
*Unbiased*

We will correct this.

*24-2*

***Why is this called a systematic error? It looks like you are merely removing the bias term in the error, which is simply a statement that the assimilation is not an unbiased estimator.***

When calculating the RMSE we remove the bias between the AR and the NR and between the CR and the NR. We then make the common equivalence between systematic error and bias. We will clarify this point in the revised version.

**25-1**
***However, both the CR and AR miss the high frequency min/max. Why?***

The AR and the CR capture the variability but not the values of the peaks. However, the LEO only samples at most twice a day over Paris and may not capture the peaks. We indicate the S5-P revisit time by the plus signs at the top of the panel and when you zoom in one sees that the peaks do not coincide with the time of the S5 P measurements. Thus, S5P cannot capture the value at the peak. Another factor could be that the emission inventory used in the AR has lower values than the one used in the NR. We will clarify this in the revised version.

**25-2**
***This is a weakness of the OSSE design, not the measurements.***

Maybe we have mis-understood the referee's comment, but our view is that because we do not know the fires a priori, we cannot include them in the CR and the AR. In our view, this result shows the benefits from the measurements regarding the identification and quantification of fire emissions.

**25-3**
***Please explain why the variability is high in Paris but not in E. Europe.***

The variability is higher over Paris than over E. Europe, because there are higher emissions over Paris than over E. Europe (as shown in Fig. 7 – old paper submission). We will clarify this in the revised version.

**25-4**
***This seems like a discovery for the authors post-facto. I recommend using this version of the OSSE only rather than devoting a whole section to it. It's what should have been done originally.***

We agree with the reviewer. However, we think it is relevant to present the results in this way because it shows the limitations of using standard operational criteria as we did in the first experiment of the OSSE. We will make this point in the revised version.

**26-1**
***The first several paragraphs are repetitious of the introduction. I recommend removing. In fact, the OSSE has shown that S5P will have a similar or bigger impact on the free trop rather than merely the surface.***

We will edit the conclusions following the reviewer's comments. The focus of this study is the surface; however, a study of the increments (see figs. 6 and 8 in the old paper submission) indicates that S5-P has benefits in the free troposphere. We will mention this in the revised version.

---

## Author Response (AR1)

**Referee # 1**

We thank the referee for his/her helpful comments. Our response to referee #1 is below. Items in bold and italics are the referee comments.

**L30: ". . .with the largest benefit occurring over land in remote regions", explain remote from what? Sources?**

*We mean regions far away from important CO sources. We rephrased this in the introduction. See line 29*

**L60-68: several sensors are listed but all references are from the MOPITT team. Please add references for the relevant sensors.**

*We added additional references in particular for AIRS and IASI CO in the revised version. See line 63*

**L81-103 paragraph: In the discussion of S-5, S-4, and S-5P, etc.., please list the timeframe of these missions. Since OMI and SCIAMACHY are discussed, as well as S-5P, this should be a good place to introduce TROPOMI. Among the sensors/missions discussed in this paragraph, which ones have CO, since it's the topic of study here?**

*We mention now the time frame for S-4. S-5 and S-5P time frames are already mentioned (see lines 88 and 93-95). We also mention TROPOMI in this paragraph but note we describe the instrument characteristics and show how the the S-5P simulated measurements are generated in section 2.2. We replace GOME-2 by IASI which measures CO and has the same revisit local time (line 98-99).*

**L99-101: "The S-5P LEO platform will address the challenge of limited revisit time from LEOs by providing unprecedented high spatial resolution of 7x7 km, and improved sensitivity in the Planetary Boundary Layer (PBL), allowing resolution of, e.g., derived CO emission sources at finer scales than hitherto." How?**

*We clarify this point in the revised version. We remove the vague sentence concerning the sources (lines 102-104)*

**L223: ". . .the NR has a realistic representation of the CO diurnal cycle." Does CO have diurnal cycle? Also, describe the ground measurement methods. Is it in situ or radiometric?**

*At air surface usually carbon monoxide exhibits a diurnal variation, generally with two peaks, one in the morning and the second in the evening. Ground station measurements are real in situ observations taken from AirBase data set. We mention this in the revised version. See line 213.*

**L232-233: ". . .the behaviour of the CO time-series from the CR compared to the NR,is similar to the behaviour of the NR CO time-series compared to the Airbase data." Not clear, are the differences similar? You might want to add the difference 100*(NRCR)/NR.**

*We likely are in the configuration where the NR is between the GS data and the CR with similar behaviour that means in our assimilation we may need a similar correction for the CR to obtain the NR or for the NR to obtain the GS. We rephrase this in the revised version. See lines 246-248*

**L237-242: TROPOMI should have been introduced in the Introduction.**

*See previous answer.*

**L308:** "Over sea, . . ." should probably be "Over the ocean, . . ."

*corrrected.*

**L387:** ". . .different to . . ." to ". . .different from . . ."

*Corrected (line 408 in the revised version)*

**L391:** ". . . the OSSE will more realistically simulate. . ." to ". . . the OSSE will simulate more realistically . . ."*

*Corrected (line 412)*

**L392:** "This follows our guiding principle to . . ." to "This allows us to . . ."

*Corrected (line 413)*

**L394-395:** "As mentioned in Sect. 2.3, we use the MOCAGE model to generate the CR. In this OSSE study, the CR is a free model run." to "In this OSSE study, the CR is a free model run using MOCAGE."

*Corrected (line 415*)

**L409:** ". . . and about 800 m in the neighbourhood of the tropopause . . ." to ". . .and approximately 800 m near the tropopause . . ."

Corrected

**L417:** . . . as they are a priori not known." To ". . . as their a priori is not known."

*We corrected by (… as their a priori is unknown). See line 439*

**L420:** ". . ., helping to differentiate the CR from the NR." To ". . ., which helps to differentiate the CR from the NR."

*Corrected (line 441)*

**L420:** "As for the NR, . . ." to "Similar to the NR, . . ."

*Corrected (line 442)*

**L435-436: add "the"**

*Corrected. See line 451*

**L453-454:** ". . . for the B-matrix: Lx and Ly are . . .; Lz is constant and . . ." to ". . .for the B-matrix, where Lx and Ly are . . .; and Lz is constant and . . ."

*Corrected. See line 474-475*

**L459:** ". . . (see Fig. 3 in Sect. 3.1)." should be Fig. 5

*Corrected. All the figure are renumbered in the revised version*

**L504: Should be Figure 5, not Figure 3**

*Corrected*

**L564-570: The reasons the AR are not performing well over fire emissions are not explained correctly in this paragraph, as suggested by L670-676. Should move part of L670-676 to this part of the paper to properly explain why AR did not work well over fires.**

*We took into account this suggestion.*

**L642: "Figure 11 shows that the AR . . ." would be better to "The AR . . ."**

*We correct this in the revised version (line 668)*

**L651: ". . . (iii) an area in the Easter part of the study domain, . . ." should name the location/country/regions.**

*We will correct this in the revised version.*

**L662-665: same issue explaining MOCAGE assimilation over fires.**

*We will rephrase this paragraph and give more details.*

**Fig. 5. Legends and labels are not legible when overlap with dashed lines. Should redo with care and use larger fonts.**

*We enlarge former Fig. 5 (now Fig. 6) but we could provide all the figures in their original format to have the best quality.*

**Referee # 2**

We thank the referee for his/her helpful comments. Our response to referee #2 is below. Items in bold and italics are the referee comments.

*While they establish a bias and variance between the NR and independent observations, their fundamental threshold is that the difference appears "reasonable".*
*What they have not done is to relate the errors between the NR and independent observations to the interpretation of the performance of the AR. So, if the accuracy and precision of the NR is twice as bad, what should one infer about the performance of the AR?*

We wrote a paragraph which clarified this point. See lines 232-237.

*I would argue that a more important implementation of their guiding principle is to assimilate real observations, e.g., MOPITT and/or SCIAMACHY, into their system and compare the analysis fields to independent observations. Then, they could do an OSSE for the same observing system and assess the statistical difference between the AR and NR sampled at independent observations versus AR(real) against independent observations. That would provide a better sense of what the OSSE limitations actually are.*

We think that the suggestion of the referee is valuable, and suitable for a study in its own right. However, it would only provide direct information on the error of the OSSE for MOPITT and/or SCIAMACHY CO and not directly for S5P. The S5P instrument has different characteristics to MOPITT and SCIAMACHY. Instead, and to keep the study tractable, we focus on the evaluation of the uncertainty in the NR, and compare it to published studies using the MOCAGE DA system. This in line with standard practice in AQ OSSEs as discussed in Timmermans et al (2015).

*As it stands, I'm still suspicious of the overall performance. Furthermore, we don't know how well S5p will perform given other sensors, e.g., MOPITT, CrIS, are already taking CO data with comparable performance.*

In this AQ OSSE, we follow standard practice by comparing the performance of S5P against a free model run. If the proposed satellite data are to have added value, they must perform better than a model. This is the first step one must take to evaluate the added value of a proposed satellite instrument. We mentioned this in the conclusion, see lines 759-769

***Otherwise, the overall work is reasonable and the authors have performed some nice statistical analysis of the results. Of course, in principle, this OSSE should have been performed \*before\* S5p was funded to assess its potential. But, practice is still catching up with theory. I've attached comments of the manuscript in the accompanying pdf.***

We thank the referee for this comment. We address below the other points from the referee.

*4-1*
*Need to include CrIS.*

Wel added the information on CrIS and the reference Fu et al., 2016, see lines 63 and 66

*4-2*
*What about planned?*

We will check if there are plans for GEO missions to measure CO. To our knowledge no GEO missions to measure specifically CO are planned yet.

*4-3*
*Why is methane defined twice?*

We identify that the formula for methane is $CH_4$.

*4-4*
*How is this point relevant if you're not discussing geostationary options?*

We think it helps the reader to contrast the characteristics of GEO and LEO satellite platforms with respect to atmospheric composition.

*5-1*
*Relative to what, MOPITT? Not clear how since they both have NIR channels.*

We mean that the S5P with its SWIR band will do better than our model in the PBL. Furthermore, compared to TIR instruments such as IASI, we expect S5P to do better in the PBL (Veefkind et al., 2012). See lines 101-104

*5-2*
*Poorly formed sentence. And not quite true. The resolution of an inverse emission estimate is controlled by the data and transport/diffusion. Not clear which is the limiting factor without analysis.*

We removed the sentence

*5-3*
*True, but how is relevant?*

We think it is helpful to remind the reader where the PBL is located.

*5-4*
*So, TROPOMI needs to be assessed relative to the existing satellites.*

See response to the general comments from this referee.

*6-1*
*So, if one knew CO concentrations or emissions better, what societal or scientific benefit would have been achieved? Improved forecasts? Better attribution?*

We provide an example of the benefits of improved knowledge of the CO distribution. See lines 140-143

*7-1*
*Shouldn't the control run include assimilation of the existing observing system, e.g., MOPITT, AIRS, CrIS, rather than a free run? This would be true for your 2003 test case when MOPITT and AIRS was available.*

See the response to the general comments of this referee.

*7-2*
*I shouldn't have to look up the figure. How does it relate to the specific OSSE elements listed?*

We provide this figure and relate it to the OSSE elements listed (this is Fig. 1 in the revised paper). We renumbered all the other figures.

*9-1*
*Did I miss something? Where is the CR described?*

We refer to section 2.4, which describes the CR. See lines 178-180

*9-2*
*There is also some high frequency component that is missed. What is that frequency? is that the night time values? Needs to be discussed.*

We discuss this high frequency component. See lines 231-236

*9-3*
*That statement needs to be limited to the ultimate performance of the OSSE. The assimilation can't saying anything better than 10-20% in accuracy.*

We rephrase this sentence following the referee's suggestion. See lines 247-248

*10-1*
*At what accuracy?*

We quote Veefkind et al (2012) on this point: 15% (accuracy) and 10% (precision). See lines 261-262 in the revised version.

*10-2*
*How low?*

We quote Veefkind et al., 2012 on this point (the value is 2%). See line 267

*10-3*
*More accurately compared to what?*

Original sentence reads:

"The use of S-5P CO total column measurements with inverse modelling techniques will also help quantify more accurately biomass burning emissions and map their spatial distribution."

We remove "more accurately"

The statement refers to the current observing system, consisting of, e.g., IASI, MOPITT, OMI, GOME-2, including measurements of the species CO and $NO_2$. S5P will provide global coverage, enhanced sensitivity for CO at the surface (compared to, e.g., IASI) and 3.5 to 7 km high spatial resolution observations.

***11-1***
***But OMI is quite a bit larger in footprint than TROPOMI. MODIS would be a better choice. Describe how the cloud fraction is related to the 7km footprint. How different are cloud ODS between the UV and the NIR? It seems like you are assuming they are the same.***

The cloud fractions were derived at the resolution of the ECMWF 0.25 x 0.25 degree grid. This is ca. 30 x 30 $km^2$ at the equator and decreases as a function of latitude. The ground pixel of OMI UV-2 and VIS channels is 13 x 24 $km^2$ at nadir increasing to 13 x 128 $km^2$ at edges of the swath. We consider that the ECMWF grid cells and OMI pixels are of comparable size for the purpose of comparing the cloud fraction distributions (ca. 0.5 million pixels or cells in each distribution). We model clouds with a simple Lambertian reflectors and ignore any wavelength dependency of cloud fraction.

We include this information in the revised version. See lines 296-301.

***12.1***
***Provides***

We think provide is appropriate

***12-2***
***Is it released now?***

Yes, it is available from the ESA Sentinel-5P TROPOMI document library:

https://sentinel.esa.int/web/sentinel/user-guides/sentinel-5p-tropomi/document-library.

We provide this information in the revised paper. See line 329

***12.3***
***sea***

Corrected
***13.1***
***Cloudy***

We understand clouded and cloudy are both appropriate here. If insisted upon, we will change this.

***13-2***
***It looks like you are assuming that the retrievals will work under partially cloudy scenes. MOPITT NIR only works under clear sky. Provide a reference on NIR CO retrievals under partially cloudy conditions and justify why a weighted approach works. Also I haven't heard any discussion of aerosols. These will be important for emissions like biomass burning and industry.***

TROPOMI NIR CO retrievals in partially cloudy conditions are discussed in Landgraf et al.: "Carbon monoxide total column retrievals from TROPOMI shortwave infrared measurements", Atmos. Meas. Tech. Discuss., doi:10.5194/amt-2016-114. They are also discussed in Vidot et al.:"Carbon monoxide from shortwave infrared reflectance measurements: A new retrieval approach for clear-sky and partially cloudy atmospheres", Remote Sens. Environ., doi:10.1016/j.rse.2011.09.032. We did not include aerosol effects in our study. We add these references to the paper. See line 354.

***13-3***

*This is the first time MOCAGE is mentioned even though it is implicitly referenced in the CR run. Needs to be mentioned earlier.*

At the start of section 2, when we first mention the CR, we will introduce MOCAGE and provide appropriate references. See lines 178-180.

*16-1*
*That is not exactly true. The existence of burning and the burnt area can be obtained from optical measurements. That is very important a priori information. Is that being ignored?*

The visible information on burnt area and burning does not provide direct knowledge on CO concentrations. It is not used in this study.

*17-1*
*In the introduction, the authors argued for the value of TROPOMI to resolve emissions. Here the focus has shifted to concentration estimation. What is the scientific rationale? What are the limitations of this OSSE, then, to make statements about resolving sources?*

We reword the text in the introduction to state that we focus on CO concentrations. We will clarify the scientific rationale of the paper and indicate the limitations of the OSSE. See lines 118-123 and 150-152.

*21-1*
*The correction in the free trop points to the role of boundary conditions in the assimilation, which would be an important consideration for a GEO. How much change is occurring at the boundary of the nested grid?*

First, note that the OSSE concerns S5-P, which is a LEO. Nevertheless, we focus on the surface level and we assume that the effect of the free troposphere on the boundary layer is secondary. Due to the revisit time of S5-P, we expect the impact at the boundaries to be small. Second, for efficiency reasons and storage limitations, we set up our DA system to only store the data over the regional domain. This means that without rerunning the OSSE, we cannot quantify the response to the reviewer's comment.

*21-2*
*That is not obvious. The biggest weakness is using 3D-var and having poor prior statistics. Given that significant fires generally last longer than a day, a proper inversion system would pick those up. That does not diminish the value of GEO sounders.*

We clarify that. We are now talking about concentrations and not emission inversions. See lines 589 and 594.

*21-3*
*Please elaborate as to whether it is the diurnal sampling or the effective sampling density that is more important for a geo.*

The comment from the referee is not clear to us. We think that diurnal sampling (high temporal resolution) will be the determining factor, as the relatively coarse model resolution would compromise the high spatial sampling.

*21-4*
*That's too qualitative. Could please provide some simple metrics, e.g., means, to quantify these statements?*

We quantify this difference in the revised paper. See lines 584-589

*23-1*
*Why?*

Owing to the relatively small variability of CO over remote land regions, the S5-P data can provide a larger benefit compared to regions where the variability is relatively high. We make this point in the revised version. See lines 651-653

*24.124*
*Unbiased*

Corrected

*24-2*
*Why is this called a systematic error? It looks like you are merely removing the bias term in the error, which is simply a statement that the assimilation is not an unbiased estimator.*

When calculating the RMSE we remove the bias between the AR and the NR and between the CR and the NR. We then make the common equivalence between systematic error and bias. We clarify this point in the revised version. We call it now bias. See lines 651-653.

*25-1*
*However, both the CR and AR miss the high frequency min/max. Why?*

The AR and the CR capture the variability but not the values of the peaks. However, the LEO only samples at most twice a day over Paris and may not capture the peaks. We indicate the S5-P revisit time by the plus signs at the top of the panel and when you zoom in one sees that the peaks do not coincide with the time of the S5 P measurements. Thus, S5P cannot capture the value at the peak. Another factor could be that the emission inventory used in the AR has lower values than the one used in the NR. We clarify this in the revised version. See lines 683-687.

*25-2*
*This is a weakness of the OSSE design, not the measurements.*

Maybe we have mis-understood the referee's comment, but our view is that because we do not know the fires a priori, we cannot include them in the CR and the AR. In our view, this result shows the benefits from the measurements regarding the identification and quantification of fire emissions.

*25-3*
*Please explain why the variability is high in Paris but not in E. Europe.*

The variability is higher over Paris than over E. Europe, because there are higher emissions over Paris than over E. Europe (as shown in Fig. 7 – old paper submission). We clarify this in the revised version. See lines 695-696.

*25-4*
*This seems like a discovery for the authors post-facto. I recommend using this version of the OSSE only rather than devoting a whole section to it. It's what should have been done originally.*

We agree with the reviewer. However, we think it is relevant to present the results in this way because it shows the limitations of using standard operational criteria as we did in the first experiment of the OSSE. We make this point in the revised version. See lines 697-698 and 726

*26-1*
*The first several paragraphs are repetitious of the introduction. I recommend removing. In fact, the OSSE has shown that S5P will have a similar or bigger impact on the free trop rather than merely the surface.*

We will edit the conclusions following the reviewer's comments. The focus of this study is the surface; however, a study of the increments (see figs. 6 and 8 in the old paper submission) indicates that S5-P has benefits in the free troposphere. We mention this in the revised version. See the conclusions.

[revised manuscript text omitted]

---

## Author Response (AR2)

Dear Editor,

Thank you for editing our paper. We took into account all technical corrections made by the reviewers and we updated accordingly the version of the paper. In addition, we provide the figures with the best quality in a separate file that could be included in the final version.
Note also, the two co-authors from ESA  (D. Schuettemeyer and B. Veihelmann) decided not to be longer among the list of authors. Therefore we removed their names from the list.

Best regards,

Rachid Abida et Jean-Luc Attié on behalf of the authors.

For your information, below are the reviewer's remarks :

Technical corrections:
1. The new sentences since last version should be English edited, from co-authors' assistance.
2. References to multiple sensors should be in chronicle orders, e.g. MOPITT, AIRS, TES, IASI, and CrIS, etc. This also apply to the years of publications (e.g., 2004, 2006, 2006, 2013, 2013, 2015, 2016).
3. The font sizes on all figures' (except Figs. 1, 3, and 14 ) labels, titles, etc. should be larger.